# `VILLA-X`: Enhancing Latent Action Modeling in Vision-Language-Action Models

**Xiaoyu Chen**[2*†], **Hangxing Wei**[3*†], **Pushi Zhang**[1*], **Chuheng Zhang**[1*], **Kaixin Wang**[1*],
**Yanjiang Guo**[2], **Rushuai Yang**[4†], **Yucen Wang**[5†], **Xinquan Xiao**[2†],
**Li Zhao**[1*§], **Jianyu Chen**[2], **Jiang Bian**[1]
[1]Microsoft Research    [2]Tsinghua University    [3]Wuhan University
[4]Hong Kong University of Science and Technology    [5]Nanjing University

## Abstract

Vision-Language-Action (VLA) models have emerged as a popular paradigm for learning robot manipulation policies that can follow language instructions and generalize to novel scenarios. Recent works have begun to explore the incorporation of *latent actions*, abstract representations of motion between two frames, into VLA pre-training. In this paper, we introduce `villa-X`, a novel Vision-Language-Latent-Action (ViLLA) framework that advances latent action modeling for learning generalizable robot manipulation policies. Our approach improves both *how latent actions are learned* and *how they are incorporated into VLA pre-training*. We demonstrate that `villa-X` can generate latent action plans in a zero-shot fashion, even for unseen embodiments and open-vocabulary symbolic understanding. This capability enables `villa-X` to achieve superior performance across diverse simulation tasks in SIMPLER and on two real-world robotic setups involving both gripper and dexterous hand manipulation. These results establish `villa-X` as a principled and scalable paradigm for learning generalizable robot manipulation policies. We believe it provides a strong foundation for future research.

## 1 Introduction

Latent action learning has emerged as a promising approach for the pretraining of vision-language-action (VLA) models (Collaboration et al., 2023; Black et al., 2024; Li et al., 2024a;c; Chen et al., 2024b; Kim et al., 2024; Zhao et al., 2025; NVIDIA et al., 2025a), enabling learning from both robot data and human video data (Ye et al., 2024; Chen et al., 2024b; NVIDIA et al., 2025a; Bu et al., 2025a). At the core of these methods is a Latent Action Model (LAM), which is designed to capture the motion semantics between successive frames into compact latent tokens. These tokens, referred to as latent actions, serve as pseudo-action labels, enriching robot policy training by enabling imitation learning on abundant, action-free data.

While promising, the central challenge still lies in improving how latent actions can enhance robot policy learning. This motivates our investigation into two pivotal questions: *how to better learn latent actions*, and *how to more effectively integrate them into VLA pre-training*? In this paper, we introduce `villa-X`, a novel Vision-Language-Latent-Action (ViLLA) framework that advances both key aspects of latent action modeling. For the latent action learning component, existing latent action models (Ye et al., 2024; Chen et al., 2024b; NVIDIA et al., 2025a; Bu et al., 2025a) typically compress latent actions based on visual signals, as shown in Figure 1 (a). However, while visual changes generally align with robot physical dynamics, certain motions, such as end-effector rotations or gripper movements, are subtle in pixel changes but critical for control. Vision-based models often pay less attention to these motions, a limitation also noted in recent work (Chen et al., 2024a). As a result, the learned latent actions remain physically ungrounded, hindering effective knowledge transfer. To address this, we move beyond purely visual signals by leveraging structural cues for physical grounding. Specifically, we integrate a proprioceptive Forward Dynamics Model (proprio FDM) as an auxiliary decoder within our Latent Action Model (LAM), as shown in Figure 1 (b). This module predicts future robot proprioceptive states and actions by including embodiment context as

---

*Equal contribution    †Interns at Microsoft Research    §Project lead

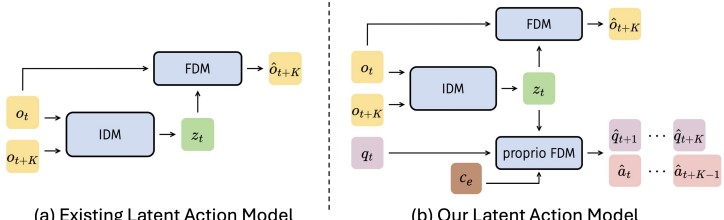

Figure 1: (a) A standard Latent Action Model (LAM) learns a latent action $z_t$ primarily through visual reconstruction, predicting a future frame $\hat{o}_{t+K}$ from the current frame $o_t$ and latent action $z_t$. (b) Our proposed model enhances this by adding a proprio-FDM. This auxiliary module predicts future robot states $\hat{q}_{t+1:t+K}$ and actions $\hat{a}_{t:t+K-1}$ conditioned on an embodiment context $c_e$, enabling the latent actions to be better grounded in physical dynamics.

inputs to help distinguish heterogeneous data. As a result, the latent actions become better grounded by focusing on visual changes that align with the agent's physical dynamics. This makes the latent action a more effective bridge between vision and control, ultimately improving knowledge transfer. This framework is general and can be extended to other cues like end-effector keypoint detection or human hand pose estimation, which we leave for future work.     To better leverage learned latent actions, we introduce a novel integration strategy for VLA pre-training. `villa-X` models latent and robot actions within a joint diffusion framework composed of two components: a latent action expert (ACT-latent) and a robot action expert (ACT-robot) as shown in Figure 2. Within this framework, an attention mechanism conditions robot action generation on latent actions generation. Compared to existing methods, this framework facilitates a more effective and structured transfer of information.

We conduct comprehensive evaluations of `villa-X` across diverse environments. Our experiments yield two key findings. First, extensive ablation studies confirm that our proposed improvements to the latent action model and policy architecture outperform existing methods. Second, we show that by scaled pre-training, the latent action expert effectively plans for the future, and generalizes zero-shot to unseen embodiments and open-vocabulary symbolic icons. Collectively, `villa-X` achieves state-of-the-art performance in various tasks including numerous simulation tasks in SIMPLER and two real-world setups featuring various robotic platforms with both gripper and dexterous hand manipulators. This establishes a robust foundation for future research in the field. In summary, our main contributions are as follows:

- We improve latent action learning by introducing an extra proprio FDM, which grounds latent actions in physical dynamics by aligning with underlying robot states and actions.

- We propose to jointly learn a latent action expert and a robot action expert through joint diffusion in the policy model, conditioning robot action prediction on latent actions to fully exploit their potential.

- Through scaled pretraining, our latent action expert develops strong zero-shot generalization capabilities. This enables the effective transfer of knowledge across diverse simulated environments and real-world robotic tasks, leading to superior performance.

## 2 RELATED WORK

**Vision-Language-Action Models**    Vision-Language-Action (VLA) models (Black et al., 2024; Li et al., 2024a; NVIDIA et al., 2025a; Chen et al., 2024b; Kim et al., 2024; Zhao et al., 2025; Li et al., 2024c; Collaboration et al., 2023) leverage pre-trained vision-language models (VLMs) to generate robot actions using visual and language cues. They either directly repurpose VLMs for action prediction (Chen et al., 2024b; Ye et al., 2024; Li et al., 2024c; Kim et al., 2024) or use action experts to map VLM outputs to robot actions (NVIDIA et al., 2025a; Black et al., 2024; Li et al., 2024a). While training on large-scale datasets like Open X-Embodiment (Collaboration et al., 2023) enhance the generalization ability of VLAs, cross-embodiment generalization remains challenging due to diverse robot setups and configurations. Utilizing unlabeled trajectory data with pseudo-labels such as latent actions (Ye et al., 2024; Chen et al., 2024b), language sub-goals (Mu et al., 2023), or visual sub-goals (Zhao et al., 2025) supports overcoming these challenges. Our method advances the latent action framework by enhancing both the modeling of latent actions and their integration into VLA pre-training.

Figure 2: **Architecture of ACT:** A hierarchical policy that predicts latent action plans and conditions robot action generation on them, incorporating embodiment context and attention masking.

**Modeling Latent Actions for VLA Pretraining** Recent exploration into latent actions began with LAPO (Schmidt & Jiang, 2023) and Genie (Bruce et al., 2024), which primarily focused on the video game domain. Dynamo (Cui et al., 2024) adopted a similar architecture, using inverse and forward dynamics models to shape state representations.

For robotic learning, methods have started to incorporate latent actions into VLA pretraining (Ye et al., 2024; Chen et al., 2024a;b; NVIDIA et al., 2025a; Bu et al., 2025a;b; Yang et al., 2025a; Liang et al., 2025). LAPA (Ye et al., 2024) proposes to learn from videos, and trains its latent actions and Vision-Language Model (VLM) using either human or robot video data. Concurrently, IGOR (Chen et al., 2024a) learns latent actions from a mixture of human and robot videos, marking the first successful transfer of latent actions between humans and robots. Moto-GPT (Chen et al., 2024b) co-fine-tunes both latent and robot action labels. GR00T (NVIDIA et al., 2025a) treats latent actions as a distinct embodiment, while Go-1 (Bu et al., 2025a) generates robot actions conditioned on discrete latent tokens. UniVLA (Bu et al., 2025b) proposes a two-stage training pipeline to learn task-centric latent actions. More recent works like Yang et al. (2025a); Liang et al. (2025) explore the continuous latent actions. Zhang et al. (2025) provides the analysis on the learned latent actions, while LAOM (Nikulin et al., 2025) uses supervision to learn latent actions that are robust to distractors in MuJoCo environments. By contrast, our approach jointly models latent and robot actions through a joint diffusion process, conditioning robot action generation on latent actions for more effective and structured information transfer. Our method improves upon prior work in several key aspects: it offers tighter integration than LAPA (Ye et al., 2024) and GR00T (NVIDIA et al., 2025a); it incorporates immediate visual context, unlike Moto-GPT (Chen et al., 2024b); and it avoids teacher-forcing inconsistencies seen in Go-1 (Bu et al., 2025a). These advantages collectively contribute to more robust reasoning at test time.

## 3 METHOD

Our method, **villa-X**, learns a physically grounded latent action space and uses it to train a VLA policy. The framework has two components:

(i) **Latent Action Model (LAM)** infers latent actions from a pair of observations, aligning these latent actions with robot dynamics via additional proprioceptive supervision.

(ii) **ACTor Module (ACT)** builds on a pre-trained vision-language backbone and jointly models sequences of latent and robot actions for improved planning and control.

Training proceeds in three stages: (i) LAM pretraining on diverse datasets, (ii) ACT pretraining with joint latent–robot modeling, and (iii) embodiment-specific finetuning.

### 3.1 LATENT ACTION MODEL (LAM)

Latent actions provide a compact intermediate representation, enabling the use of abundant human videos and improving cross-embodiment generalization (Ye et al., 2024; Chen et al., 2024b). Prior works typically learn a quantized codebook of latent actions via two modules: an *Inverse Dynamics Model* (IDM) and a visual *Forward Dynamics Model* (FDM). The IDM predicts a latent token $z_t$ from a frame pair $(o_t, o_{t+K})$, while the FDM reconstructs the future observation $\hat{o}_{t+K}$ from $(o_t, z_t)$:

$$z_t = \text{IDM}(o_t, o_{t+K}), \quad \hat{o}_{t+K} = \text{FDM}(o_t, z_t). \tag{1}$$

This objective ensures consistency in visual change but ignores physical dynamics, producing latents that are insufficiently grounded when robot states are available.

**Proprioceptive Grounding.** To address this, we introduce an additional *proprioceptive Forward Dynamics Model (proprio-FDM)* that predicts both future robot states and actions $K$ steps ahead, given the current state $q_t$ and latent $z_t$:

$$(\hat{q}_{t+1}, ..., \hat{q}_{t+K}, \hat{a}_{t+1}, ..., \hat{a}_{t+K}) = \text{proprio-FDM}(q_t, z_t, c_e), \tag{2}$$

where $c_e$ denotes an embodiment context described below. Optimizing visual and proprioceptive forecasting jointly encourages latent tokens to emphasize physical dynamics alongside visual changes.

**Disambiguating Heterogeneous Embodiments.** Large-scale datasets mix embodiments with different morphologies and control frequencies. Naively conditioning the proprio-FDM on $(q_t, z_t)$ risks encoding embodiment-specific features into latents. We introduce a context vector $c_e$ comprising:

$$c_e = f(\text{dataset ID, control frequency}), \tag{3}$$

where dataset IDs are mapped to learnable embeddings, and frequencies are encoded using sinusoidal features passed through an MLP. These embeddings, concatenated with $q_t$, allow proprio-FDM to separate embodiment-specific dynamics while preserving latent action consistency across datasets.

The full LAM thus optimizes image reconstruction loss, proprioceptive prediction loss, and vector-quantization commitments jointly. For human video lacking proprio labels, the proprio term is omitted. Finally, we adopt the continuous vector from the VQ codebook center as our latent actions. We refer readers to Appendix A for further implementation details.

In summary, our LAM extends prior latent action models beyond compressing only visual changes to jointly modeling physical state transitions. While this work leverages robot proprioception for grounding, the framework is generic: alternative structural cues like end-effector keypoint detection or human hand pose estimation could replace low-level states, which we leave for future exploration.

## 3.2 ACTOR MODULE (ACT)

Our `ACT` module extends traditional vision-language-action (VLA) approaches by explicitly modeling both *latent actions* ($z_{t:t+(n-1)K}^K = (z_t, z_{t+K}..., z_{t+(n-1)K})$) and *robot actions* ($a_{t:t+m-1} = (a_t, a_{t+1}, ..., a_{t+m-1})$) with a joint diffusion process. We factorize the policy into two conditional distributions:

$$\pi(a_{t:t+m-1}, z_{t:t+(n-1)K}^K \mid o_t, l, q_t, c_e) = \underbrace{\pi_{\text{robot}}(a_{t:t+m-1} \mid z_{t:t+(n-1)K}^K, o_t, l, q_t, c_e)}_{\text{ACT-robot}}$$
$$\cdot \underbrace{\pi_{\text{latent}}(z_{t:t+(n-1)K}^K \mid o_t, l)}_{\text{ACT-latent}}. \tag{4}$$

where $o_t$ is the observation, $l$ the language instruction, $q_t$ the proprioceptive state, and $c_e$ the embodiment context. Additionally, the low-level policy can optionally incorporate wrist camera input if available. This explicit modeling and factorization improves upon prior methods, such as LAPA (Ye et al., 2024), which rely on latent actions only through pre-trained weight initialization. By contrast, our method treats latent actions as a distinct mid-level representation that bridges high-level vision and language prompts with low-level robot actions, and allow for allowing more effective and structured information transfer from latent actions to robot actions.

**Architecture.** `ACT` (Figure 2) comprises three experts with a blockwise causal attention mask:

- **VLM:** Encodes the visual-language inputs into high-level features.
- **ACT-latent:** Latent action expert that predicts latent action tokens for mid-level planning, conditioning on VLM features.
- **ACT-robot:** Robot action expert that produces the low-level action chunk, conditioning on VLM features, predicted latents and additional control-specific inputs including proprioceptive states and embodiment context.

**Attention Masking Strategies.** A key aspect of `ACT` is how we maintain a robust dependence on the latent tokens without letting the policy learn trivial shortcuts. Inspired by Moto (Chen et al., 2024b) and RDT (Liu et al., 2024), we adopted the stochastic Masking strategy. During training, we stochastically mask the attention from robot actions to latent actions. In 50% of the cases, *all* robot-to-latent attention are masked; otherwise, 50% of the latent tokens are randomly masked. This can prevent the robot-action branch from overly relying on latent actions and leading to improved robustness. We found this design crucial in practice.

**Joint Diffusion Modeling.** For implementation, **villa-X** models the joint distribution of the future robot actions $a_{t:t+m-1}$ and latent actions $z_{t:t+(n-1)K}^K$ using a conditional flow matching framework. For notational simplicity, we group these actions into a single variable $x_t$ and denote the conditioning inputs $(o_t, l, q_t, c_e)$ as $O_t$. The objective is to train a network $v_\tau^\theta$ that minimizes the flow matching loss:

$$L_\tau(\theta) = \mathbb{E}_{p(x_t|O_t),\, q(x_t^\tau|x_t)} \left\| v_\tau^\theta(x_t^\tau, O_t) - u(x_t^\tau \mid x_t) \right\|^2 \tag{5}$$

where $\tau \in [0,1]$ denotes flow matching timestep. In practice, we first sample random noise $\epsilon \sim N(0, I)$ to create a noisy target $x_t^\tau = \tau x_t + (1 - \tau)\epsilon$. The network $v_\tau^\theta(x_t^\tau, O_t)$ is then trained to predicted the denoising vector field $u(x_t^\tau \mid x_t) = \epsilon - x_t$. During training, we sample $\tau$ from beta distributions. Notably, the explicit factorization in Eq. 4 is achieved by block-wise causal attention.

## 4 EXPERIMENTS

In this section, we aim to answer the following questions through experiments:

- Does our improved `LAM` learn higher-quality latent actions?
- Can the actor module effectively leverage the pre-trained latent actions?
- By scaling pre-training, can the latent actor module effectively plans for the future and generalize zero-shot to unseen embodiments and open-vocabulary concepts in symbolic icons?
- How does **villa-X** compare to existing VLA baselines in both simulated benchmarks and real-world robot tasks?

### 4.1 DOES OUR IMPROVED LAM LEARN HIGHER-QUALITY LATENT ACTIONS?

In this subsection, we evaluate whether our improved latent action modeling enhances the quality of the learned latent actions. The key component of our `LAM` is the incorporation of the proprio FDM module. To assess its impact, we compare our model (denoted `w/pp`) to a variant without the proprio FDM module (denoted `wo/pp`).

**Probing** First, a core expectation for latent actions is that they should carry information useful for predicting low-level robot actions. To test this, we conduct a probing experiment. Specifically, after training the latent action models, we freeze them and train a simple 3-layer MLP to predict the corresponding robot actions for each latent action. Probing is conducted on the LIBERO dataset (Liu et al., 2023a), which is not used for training latent action models. We train the MLP on the training split of LIBERO and evaluate it using the L1 loss on the validation split.

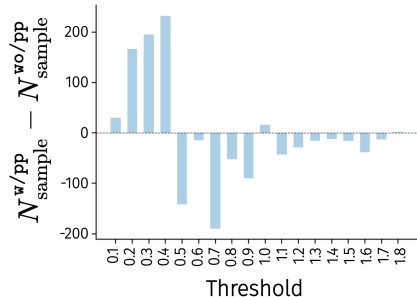

Figure 3: Probing experiment results.

We are interested in how closely the predicted actions match the ground truth. In LIBERO, the robot action space has eight dimensions: three for position, four for rotation, and one for the gripper. Rather than averaging the error across dimensions, we focus on the maximum L1 error across all action dimensions, as we want to avoid large deviations in any single aspect of the action. For each model variant (`w/pp` and `wo/pp`), we compute the number of validation samples whose maximum L1 error falls below a threshold. By sweeping this threshold, we count how many samples fall within each error bin. A better model should yield more samples with low errors.

Table 1: Evaluation results on SIMPLER for different variants of our **villa-X**(top group) and alternative approaches for incorporating latent actions (bottom group). "Ours" refers to the `w/pp` described in the main text.

| Method | Google robot | | | | WidowX robot | | | | |
|---|---|---|---|---|---|---|---|---|---|
| | Pick | Move | Drawer | Avg. | Carrot | Eggplant | Spoon | Cube | Avg. |
| Ours | **81.7** | **55.4** | 38.4 | **58.5** | 24.2 | 71.7 | **48.3** | **19.2** | **40.8** |
| `wo/pp` | 77.0 | 52.7 | **42.6** | 57.4 | 22.5 | 57.5 | 43.3 | 5.9 | 32.3 |
| `wo/LAM` | 42.1 | 24.6 | 38.4 | 35.0 | **25.8** | 60.8 | 36.7 | 9.2 | 33.1 |
| LAPA-style | 64.7 | 28.8 | 38.0 | 43.8 | 0.8 | 0.0 | 2.5 | 0.8 | 1.0 |
| Go-1-style | 44.3 | 46.3 | 41.2 | 43.9 | 7.5 | **91.3** | 30.4 | 16.7 | 36.5 |

For each error bin, we compute the difference in the number of samples between the `w/pp` and `wo/pp` variants and present the results in Figure 3. The `w/pp` variant produces more samples with smaller errors, while the `wo/pp` variant has more samples in the high-error bins. This demonstrates the effectiveness of the proprio FDM module in capturing information from the robot actions. We further visualize the learned latent actions, and perform more ablations on LAM. Please refer to Appendix D for more details.

**Policy Pre-training** Next, we compare how the latent actions generated by different variants of `LAM`(`w/pp` and `wo/pp`) influence policy pre-training. Unlike the main experiments, we pre-train models in this section on a mixture of 10% Fractal (Brohan et al., 2022) data, 10% Bridge V2 (Ebert et al., 2021) data, and 100% Something-Something V2 (Goyal et al., 2017a) data, to reduce computation cost while remaining a setting where limited robot data are available for training the VLA model. The resulting policies are evaluated in the SIMPLER environment (Li et al., 2024d), a simulation benchmark explicitly designed to mitigate the gap between simulated and real-world robotic environments. It comprises two platforms: the Google robot with three manipulation tasks and the WidowX robot with four. We evaluate our method on the visual matching setting. The results are summarized in Table 1. We observe that `w/pp` clearly outperforms `wo/pp`, demonstrating the effectiveness of incorporating the proprio FDM module. Additionally, we include a baseline that does not use latent actions (denoted `wo/LAM`) and is trained solely to predict robot actions. The performance of `wo/LAM` is significantly worse, indicating that pre-training with latent actions is essential.

## 4.2 CAN THE ACTOR MODULE EFFECTIVELY LEVERAGE THE PRE-TRAINED LATENT ACTIONS?

Given high-quality latent actions produced by the pre-trained `LAM`, we investigate whether our design can effectively leverage them to pre-train robot control policies. We compare our approach against two recent methods that also utilize latent actions, albeit in different ways: LAPA (Ye et al., 2024) and GO-1 (Bu et al., 2025a).

To isolate the effect of how latent actions are incorporated, we implement LAPA-style and GO-1-style models based on our architecture for a fair comparison. For the LAPA-style model, we follow a two-stage pre-training protocol: we first train the VLM to predict latent actions, then replace the latent action prediction head with a robot action prediction head and continue training on data with robot action labels. For the GO-1-style model, we implement a separate latent planner that autoregressively predicts latent actions. The robot action prediction component remains largely unchanged as in our main design.

Following the experiment setup in the previous subsection, we train all models on the same dataset mixture and then evaluate the resulting policies in the SIMPLER environment (Li et al., 2024d). The results are shown in Table 1. Compared to other two approaches, our method achieves significantly higher performance, validating the effectiveness of our design for incorporating latent actions into VLA pre-training. More ablation studies on policy designs can be found in Appendix F.

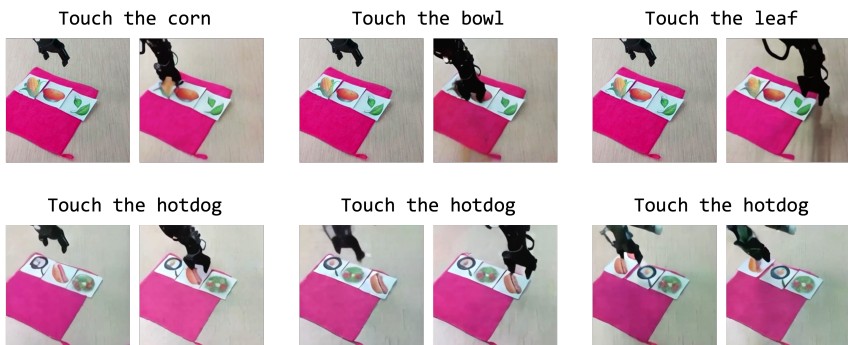

Figure 4: Visualization of zero-shot latent plans on an unseen embodiment. Each pair of images shows the starting frame (left) and the ending frame (right), with the instruction displayed above.

### 4.3 Zero-shot Generalization for Latent Actor Module

To evaluate the zero-shot generalization capabilities of ACT-latent in planning, we conducted a real-world visualization experiment focused on its ability to handle new embodiments and understand novel open-vocabulary symbols. For embodiment generalization, we used a Realman robot arm, a new embodiment **never** seen during training. To assess open-vocabulary generalization, we designed a set of symbol cards, testing the model's ability to comprehend concepts typically absent from standard robotics datasets.

The evaluation process is as follows: given a starting image and a language instruction (e.g., "touch the corn"), ACT-latent first generates a sequence of latent actions. Then a separately trained world model is used to render this sequence into a video, allowing us to verify the effectiveness of the plan.

As shown in Figure 4, the rendered trajectories confirm that the model successfully generates latent plans that follow the commands. These results highlight two key capabilities of our approach:

- Embodiment Generalization: ACT-latent successfully identifies and controls this unseen robot arm, indicating that its learned knowledge is embodiment-agnostic and readily transferable to the new robotic platform.

- Open-Vocabulary Understanding: The model's ability to interact correctly with symbol concepts reveals that **villa-X** retains the general-purpose vision-language capabilities of the original Vision-Language Model (VLM) after pre-training.

More visualizations can be found in Appendix E. To evaluate how effectively **villa-X** uses this knowledge, we measure its success rates on a variety of control tasks in the following sections.

### 4.4 Evaluating **villa-X** in Simulation

**Baselines and Experimental Setup** We use the SIMPLER benchmark as described above. In this section, we compare against several categories of prior work:

- Vision-Language-Action (VLA) models: RT-1-X (Collaboration et al., 2023), Octo-base (Octo Model Team et al., 2024), OpenVLA (Kim et al., 2024), RoboVLMs (Li et al., 2024b), $\pi_0$ (Black et al., 2024), $\pi_0$-FAST (Pertsch et al., 2025), OpenVLA-OFT (Kim et al., 2025), which learn policies solely from mixed robot datasets.

- Joint policy learning and world modeling method: GR00T-N1.5 (NVIDIA et al., 2025b), which aligns the model with target future embeddings.

- Visual trace methods: TraceVLA (Zheng et al., 2024), Magma (Yang et al., 2025b), which learn planning on the extract visual traces from videos.

- Latent-Action based methods: MoTo (Chen et al., 2024b) and LAPA (Ye et al., 2024), which additionally exploit unlabelled videos by inferring latent actions.

Table 2: Comparison on SIMPLER of **villa-X** and existing methods. Methods marked with ∗ are evaluated directly after pretraining, whereas other methods are evaluated after post-training on corresponding dataset.

| Method | Google Robot | | | | WidowX Robot | | | | |
|---|---|---|---|---|---|---|---|---|---|
| | Pick | Move | Drawer | Avg. | Carrot | Eggplant | Spoon | Cube | Avg.. |
| RT-1-X ∗ | 56.7 | 31.7 | 59.7 | 49.4 | 4.2 | 0.0 | 0.0 | 0.0 | 1.1 |
| Octo-base ∗ | 17.0 | 4.2 | 22.7 | 14.6 | 8.3 | 43.1 | 12.5 | 0.0 | 16.0 |
| OpenVLA ∗ | 16.3 | 46.2 | 35.6 | 32.7 | 0.0 | 4.1 | 0.0 | 0.0 | 1.0 |
| RoboVLMs ∗ | 72.7 | 66.3 | 26.8 | 55.3 | 25.0 | 0.0 | 20.8 | 8.3 | 13.5 |
| RoboVLMs | 77.3 | 61.7 | 43.5 | 60.8 | 20.8 | 79.2 | 45.8 | 4.2 | 37.5 |
| $\pi_0$ | 72.7 | 65.3 | 38.3 | 58.7 | 0.0 | 62.5 | 29.1 | 16.6 | 27.1 |
| $\pi_0$-FAST | 75.3 | 67.5 | 42.9 | 61.9 | 21.9 | 66.6 | 29.1 | 10.8 | 32.1 |
| OpenVLA-OFT | 72.3 | 69.6 | 47.2 | 63.0 | 4.2 | N/A | 12.5 | 8.3 | N/A |
| GR00T-N1.5 | 69.3 | 68.7 | 35.8 | 57.9 | **54.3** | 61.3 | 75.3 | 57.0 | 62.0 |
| TraceVLA | 45.0 | 63.8 | **63.1** | 57.3 | 16.6 | 65.0 | 12.5 | 16.6 | 27.7 |
| Magma | 75.0 | 53.0 | 58.9 | 62.3 | 29.2 | **91.7** | 37.5 | 20.8 | 44.8 |
| MoTo | 74.0 | 60.4 | 43.1 | 59.2 | N/A | N/A | N/A | N/A | N/A |
| LAPA | N/A | N/A | N/A | N/A | 45.8 | 58.3 | 70.8 | 54.2 | 57.3 |
| **Ours** w/o latent | 56.3 | 25.8 | 27.3 | 36.5 | 31.3 | 74.6 | 61.7 | 28.3 | 49.0 |
| **Ours** | **98.7** | **75.0** | 59.3 | **77.7** | 46.3 | 64.6 | **77.9** | **61.3** | **62.5** |

Except where noted (∗), all models follow a two-stage pretraining–finetuning protocol, including a general pretraining phase on large-scale mixed data, followed by finetuning on a dataset of specific embodiment. We also include an ablation (Ours w/o latent) that removes our latent-action expert while keeping all other components unchanged. Baseline scores are cited from their original publications or other relevant literature, while missing entries are marked as N/A.

**Experimental Results**    Table 2 summarizes the success rates on both platforms. Our full model achieves the highest score on average success rate on the Google robot (77.7%) and the WidowX robot (62.5%). This improvement over VLA methods, which cannot exploit unlabelled video, demonstrates the benefit of incorporating human videos into policy learning. Moreover, our approach outperforms other video learning and latent-action methods, indicating that our specific mechanism for leveraging video data is more effective. Finally, the gap between our full model and the "**villa-X** w/o latent" ablation confirms that the latent-action expert is essential for achieving these gains.

## 4.5    EVALUATING **VILLA-X** ON REAL-WORLD ROBOTS

To assess real-world generalization, we deploy **villa-X** on two platforms: a Realman arm with a gripper and an XArm with a 12-DoF XHand as shown in Figure 5.

**Realman robot arm with gripper**    We use a 6-DoF Realman RM 75 with a 1-DoF Inspire gripper, fine-tuning and evaluating on five tasks: Pick-in (pick the block into a bowl), Pick-out (pick the block out of a bowl), Stack (stack the block onto another block), Unstack (unstack the block from another block), and Push (push the block to a given location). The fine-tuning set contains 375 teleoperated trajectories (75 per task); the object layout and table are fixed, while object positions vary.

We conduct two sets of evaluation: In task evaluation, we remain the table setup the same as data collection; in generalization evaluation, we change the color of the block and table cover. For each task, we conduct 10 trials with distinct object positions; positions and lighting are identical across policies. As shown in Table 4, **villa-X** outperforms all baselines in both settings. Rollout videos are available in the Appendix.

**Xarm robot arm with Xhand dexterous hand**    On the dexterous-hand platform, we use the Xhand, a 12-DoF dexterous hand with five flexible fingers, mounted on a 7-DoF Xarm robot arm. Fine-tuning is performed on the Xhand Dataset (Hu et al., 2024), which comprises 4,000 trajectories spanning

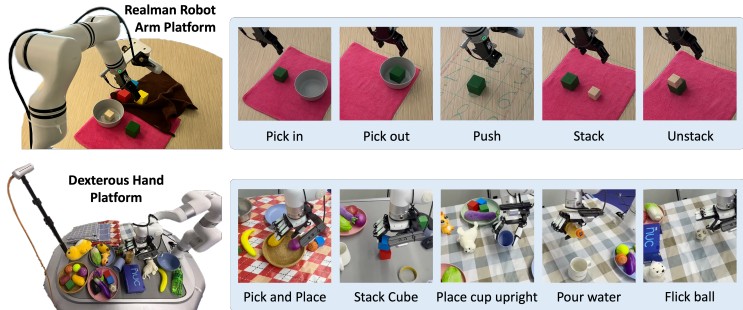

Figure 5: Real-world robot evaluation platforms: **(top)** Realman robot arm platform with a gripper and **(bottom)** Xarm robot arm with Xhand dexterous hand. Platform setups are shown on the left, with corresponding evaluation tasks on the right.

Table 3: Evaluation on Xarm robot arm of `villa-X` and existing methods.

| Method | Pick & Place | | Stack Cube | | Place Cup Upright | | Pour Water | | Flick Ball | |
|---|---|---|---|---|---|---|---|---|---|---|
| | seen | unseen | seen | unseen | seen | unseen | seen | unseen | seen | unseen |
| GR-1 | 56 | 40 | 15 | 5 | 0 | 0 | 0 | 0 | 40 | 10 |
| GR00T | 44 | 28 | 20 | 0 | 20 | 0 | 0 | 0 | 30 | 0 |
| **Ours** w/o latent | 72 | 60 | 70 | 40 | 40 | 30 | 40 | 10 | 50 | 30 |
| **Ours** | **84** | **68** | **75** | **50** | **60** | **30** | **60** | **30** | **50** | **40** |

Table 4: Evaluation on Realman robot arm of `villa-X` and existing methods.

| Method | Pick in | Pick out | Push | Stack | Unstack | Change block color | Change table cover |
|---|---|---|---|---|---|---|---|
| $\pi_0$ | 40 | 70 | **60** | 50 | 70 | 40 | 40 |
| GR00T | 30 | 70 | 10 | 10 | 60 | 50 | 30 |
| OpenVLA-OFT | **50** | 70 | 10 | 50 | 80 | 40 | 50 |
| **Ours** w/o latent | 40 | 80 | 30 | **60** | 70 | 40 | 30 |
| **Ours** | 30 | **100** | 50 | 50 | **100** | **60** | **60** |

13 task categories. Since no dexterous-hand data were used during pretraining, this evaluation can test embodiment transfer ability. We select five representative tasks—pick-and-place, cube stacking, cup upright placement, water pouring and ball flicking. The results are summarized in Table 3 for (i) seen tasks, where objects are randomly replaced or additional distractors are added, and (ii) unseen tasks, which use unseen objects or backgrounds. The performances are evaluated under 50 runs for pick and place, 20 runs for stack cube and 10 runs for others. Table 3 demonstrates that our method outperforms existing baselines.

## 5 CONCLUSION, LIMITATIONS, AND FUTURE WORKS

In this paper, we presented villa-X, a novel Visual-Language-Latent-Action (ViLLA) framework that improves both the learning of latent actions and their incorporation into VLA pre-training. Our experiments demonstrate that our enhanced Latent Action Model learns higher-quality latent actions, and our improved policy model more effectively leverages these learned latents. The learned latent action expert can even generalize zero-shot to an unseen embodiment, showing strong generalization ability. Overall, our method exhibits superior performance in both simulated environments and real-world robotic tasks. One limitation is that the proposed latent expert, although effective at future planning through both visual and proprioceptive state planning, is not fully explored in this work. For example, future research could learn a critic with prior knowledge from foundation vision-language models, allowing multiple samples from the latent expert and rejecting planned trajectories that do not follow the language instruction. We leave this aspect as future work to further improve the capability of the ViLLA framework.

## 6 REPRODUCIBILITY STATEMENT

To ensure the reproducibility of our research, we provide comprehensive details of our methodology and implementation. A thorough description of our model architecture, experimental setup, and evaluation protocols can be found in Section 3. Additional implementation details required to reproduce our main results are documented in Appendix. The source code is included in the supplementary material.

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

## A  Additional Implementation Details for LAM

In this appendix, we provide extended information on the architecture, training protocols, and inference behavior for our Latent Action Model (LAM).

### A.1  Architecture Overview

Our LAM comprises four main modules:

   (i) **Spatial–Temporal Transformer (ST-Transformer) Inverse Dynamics Model (IDM):** Takes a video clip (by default, $8 \times 224 \times 224$) as input. We employ patch embedding with a patch size of 14 and stack 12 ST-blocks (Xu et al., 2020), each with a hidden dimension of 768 and 32 attention heads.

   (ii) **Vector Quantization (VQ) Module:** Maps the continuous IDM outputs to discrete latent tokens, each associated with a codebook entry. We set the codebook size to 32. While the model internally uses discrete token indices during training, the continuous codebook centers are used in downstream modules.

(iii) **Image Reconstruction Forward Dynamics Model (FDM):** A 12-layer Vision Transformer (ViT)-base network that takes the current frame $o_t$ and a latent action $z_t$ to predict $\hat{o}_{t+K}$.

(iv) **Proprioceptive Forward Dynamics Model (proprio-FDM):** A 2-layer MLP with dual output heads to predict future robot states $\hat{q}_{t+i}$ and low-level robot actions $\hat{a}_{t+i}$. This module takes the current robot state $q_t$, the latent $z_t$, and an embodiment context vector $\mathbf{c}_e$.

Rather than predicting a single latent action per pair of frames, the ST-Transformer-based IDM processes a sequence of $T_{\text{LAM}}$ frames, resulting in $T_{\text{LAM}} - 1$ latent tokens. We use $T_{\text{LAM}} = 8$. By reconstructing future frames with the FDM and future states/actions with the proprio-FDM, the model learns a latent representation that is both visually and physically grounded.

## A.2 Training Details

We train our LAM on a combination of human egocentric data (e.g., Ego4D (Grauman et al., 2022a)) and robot trajectories (e.g., OpenX (Collaboration et al., 2023)). Samples lacking low-level robot annotations (e.g., human videos) exclude the proprio-FDM branch, using only the visual FDM objective.

**Hyperparameters.** We use a batch size of $512$ and a learning rate of $1.5 \times 10^{-4}$, with a 2000-step linear warmup. Training lasts approximately 4 days on 128 NVIDIA A100 GPUs. Both the visual FDM and proprio-FDM share the same weighting in the overall loss. Throughout training, each latent token is discretized via the VQ module but is represented by its continuous codebook center in subsequent network components.

## A.3 Inference Behavior and Diagnostics

During inference, only the IDM is required to extract latent tokens from consecutive frames. The FDM and proprio-FDM are typically retained for diagnostic and visualization purposes, allowing us to examine whether the learned latent tokens accurately capture future frame content, robot states, and actions. This reconstruction-based analysis aids in understanding and debugging the physical grounding of the latent representation.

# B Additional Implementation Details for Actor Module

Our VLA model comprises three components. First, the vision–language encoder is based on PaliGemma (Beyer et al., 2024), a 3B-parameter VLM pretrained with $224 \times 224$ images and 128-token text inputs. Second and third, the latent-action expert and the robot-action expert are each implemented as 18-layer Transformer networks, mirroring PaliGemma's design, with a hidden dimension of 1,024 and 8 attention heads. For the latent action sequence, we select a sequence length of $N = 6$, and for the robot actions, we select a sequence length of $M = 4$.

We extend our policy head with a variant of HPT (Wang et al., 2024b), assigning each embodiment its own pair of state- and action-projection layers while sharing all other parameters. Visual features from the wrist camera are extracted by a pretrained ResNet-18 (He et al., 2015) and fused into the main model via a shared cross-attention head that maps the ResNet features into 16 tokens. During training, wrist-view inputs are randomly masked 50% of the time. We also observed that the latent-action representation can be overly exploited by the robot-action expert, so we regularize this with two complementary dropout schemes. First, we add a 50% attention-weight dropout on the latent-action stream. For the remaining tokens, we randomly mask 50% latent action tokens. This combined masking strategy encourages the model to learn robust, generalizable policy that will balance the predicted latent actions as well as the input image and instruction. During training, we sample $\tau$ from different beta distributions for latent actions and robot actions, which biases the timesteps for latent actions towards the noisier regime. Each expert contains approximately 300 M parameters and is trained from scratch. We train all components jointly using a learning rate of $5e - 5$ with a 200-step linear warmup. We clip gradients to a maximum norm of 1.0 to ensure stable optimization. The pretraining takes 4 days on 64 NVIDIA A100 GPUs.

## C   DATASET

### C.1   DATA MIXTURE

We curated a data mixture by combining both robot data and action-free human videos for our pretraining phase. For robot data, we draw primarily from OpenX (Collaboration et al., 2023) mixture and AgiBot (Bu et al., 2025a). For OpenX dataset, our base data mixture is created primarily based on (Kim et al., 2024; Octo Model Team et al., 2024). In total, we use 1.6M trajectories with 223.5M frames of robot data. For human videos, we use a mixture of Ego4D (Grauman et al., 2022a), EgoPAT3D (Li et al., 2022), EGTEA Gaze+ (Li et al., 2018), EPIC-KITCHENS (Damen et al., 2020), HO-Cap (Wang et al., 2024a), HOI4D (Liu et al., 2022), HoloAssist (Wang et al., 2023), RH20T (Fang et al., 2023), Something Something V2 (Goyal et al., 2017b). Altogether, this yields 3.6M clips of human videos. During LAM pretraining, we exclusively utilize the primary third-person camera view. For policy pretraining, we optionally incorporate the wrist-mounted view (when available), applying a 50% dropout. A full breakdown of our data mixture is listed in Table 5.

### C.2   DATA PREPROCESSING

For data cleaning, we adopt EgoHOD (Pei et al., 2025), a curated subset of Ego4D (Grauman et al., 2022a), and further filter the videos based on visual quality to ensure high-quality inputs for training. For both robot data and human videos, we apply random adjustments to brightness, contrast, saturation, and hue as data augmentation. In the case of robot data, we represent both proprioceptive states and actions using euler angles.

| Dataset | Mix Ratio (%) |
|---|---|
| RT-1 Robot Action (Brohan et al., 2022) | 9.70 |
| AgiBot World Beta (Bu et al., 2025a) | 20.0 |
| Kuka (Kalashnikov et al., 2018) | 1.97 |
| Bridge (Walke et al., 2023; Ebert et al., 2021) | 5.47 |
| Taco Play (Rosete-Beas et al., 2022; Mees et al., 2023) | 0.76 |
| Jaco Play (Dass et al., 2023) | 0.12 |
| Berkely Autolab UR5 (Chen et al.) | 0.31 |
| Language Table (Lynch et al., 2023) | 0.11 |
| Stanford Hydra Dataset (Belkhale et al., 2023) | 1.61 |
| NYU Franka Play Dataset (Cui et al., 2022) | 0.22 |
| Furniture Bench Dataset (Heo et al., 2023) | 0.63 |
| Austin Sailor Dataset (Nasiriany et al., 2022) | 0.57 |
| Austin Sirius Dataset (Liu et al., 2023b) | 0.45 |
| BC-Z (Jang et al., 2022) | 3.47 |
| DLR EDAN Shared Control (Quere et al., 2020) | 0.01 |
| CMU Stretch (Mendonca et al., 2023) | 0.04 |
| FMB Dataset (Luo et al., 2024) | 0.73 |
| DobbE (Shafiullah et al., 2023) | 0.37 |
| DROID (Khazatsky et al., 2024) | 3.46 |
| Ego4D (Grauman et al., 2022b; Pei et al., 2025) | 21.46 |
| EgoPAT3D (Li et al., 2022) | 0.94 |
| EGTEA Gaze+ (Li et al., 2018) | 0.89 |
| EPIC-KITCHENS (Damen et al., 2020) | 6.95 |
| HO-Cap (Wang et al., 2024a) | 0.63 |
| HOI4D (Liu et al., 2022) | 1.99 |
| HoloAssist (Wang et al., 2023) | 4.77 |
| RH20T (Fang et al., 2023) | 5.56 |
| Something-Something V2 (Goyal et al., 2017a) | 6.82 |

Table 5: Our training data mixture used during the pretraining phase.

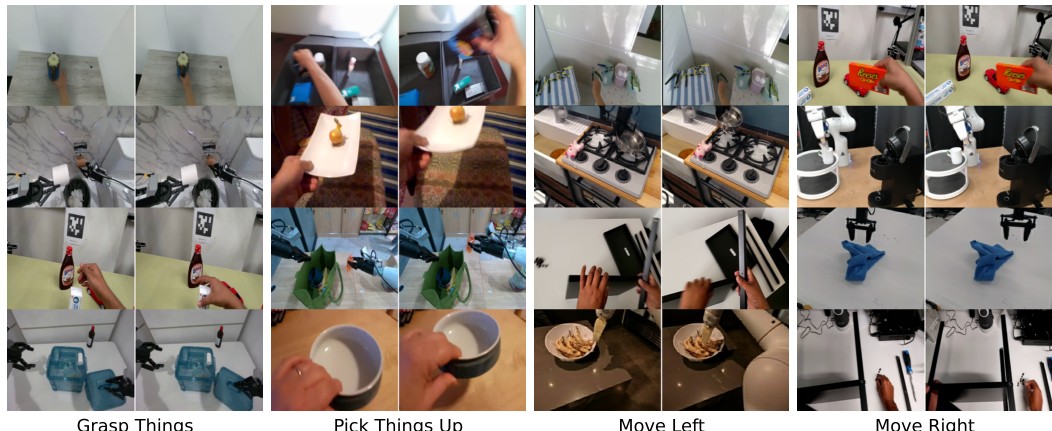

| Grasp Things | Pick Things Up | Move Left | Move Right |

Figure 6: Visualization of image pairs with similar latent actions.

# D  LAM VISUALIZATION AND MORE ABLATIONS

## D.1  IMAGE PAIRS WITH SIMILAR LATENT ACTIONS

Figure 6 visualizes image pairs sharing the same latent action, demonstrating that these pairs correspond to similar underlying robot behaviors.

The results demonstrate that similar latent actions represent the similar robot behaviors and low-level actions, in regardless of which embodiment (including human and different robots) is executing such action. This results support that **villa-X** learns cross-embodiment prior knowledge for manipulations with latent actions.

## D.2  TRANSFER VIDEO DEMONSTRATIONS INTO ROBOT ACTIONS THROUGH LAM AND PROPRIO FDM

To further demonstrate the transfer ability of our LAM, we extract latent actions from videos of task demonstrations, map them to robot actions using the proprio FDM, and execute the resulting robot actions in the SIMPLER simulator.

The results are presented in Figure 7 and Figure 8. In each figure, the top row shows the video demonstrations used by LAM to extract latent actions, while the bottom row displays the corresponding SIMPLER simulation results, where real actions decoded from the latent actions using proprioceptive FDM are executed. Specifically, Figure 7 illustrates robot-to-robot transfer, and Figure 8 illustrates human-to-robot transfer. The simulated motions closely reproduce the original demonstrations, indicating that latent actions learned by **villa-X** are both aligned with and grounded in the robot's actions.

## D.3  MORE ABLATIONS ON LAM

To validate the contribution of the embodiment context in our proprio-FDM, we further conducted an ablation study comparing our full method ("Ours") against a version without the context ("Ours w/o context"). Both models were trained on 10% of the OXE dataset.

(1) Performance on validation dataset: We measured the reconstruction loss of visual FDM and proprio FDM on the validation set:

(2) Zero-Shot Generalization to a Novel Embodiment: We evaluated the model on our dataset collected on our Realman robot arm dataset (from Section 4.4), an embodiment completely unseen during training. We then conducted the action probing experiment described in Section 4.1 by inferring latent actions with IDM and training a new MLP to predict robot actions from the latent actions.

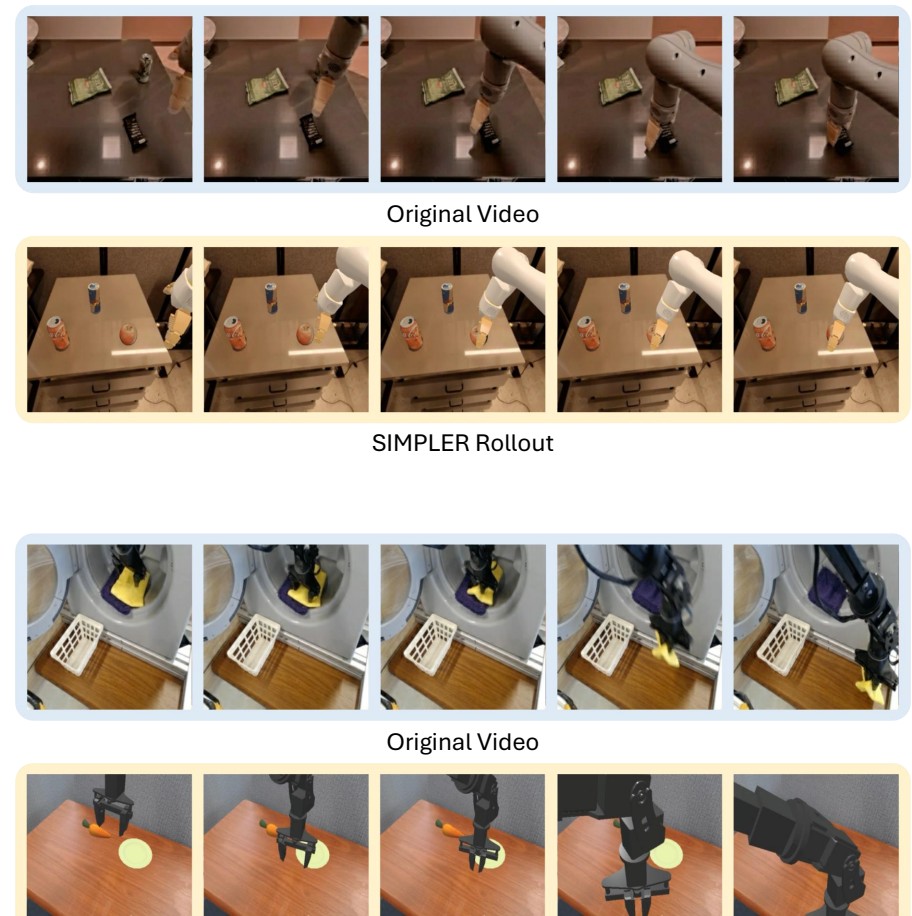

Figure 7: Transfer robot video demonstrations into robot actions through `LAM` and proprio FDM in SIMPLER simulator. Upper: the SIMPLER rollout closely reproduce the motion of moving downwards, Bottom: the SIMPLER rollout closely reproduce the motion of moving right.

| Method | Visual FDM loss ($\downarrow$) | Proprio FDM loss ($\downarrow$) |
|---|---|---|
| Ours w/o context | 0.068 | 0.078 |
| Ours | **0.057** | **0.070** |
| Relative improvement | 16.2% | 10.3% |

Table 6: Performance comparison on the validation set.

| Method | Probing loss ($\downarrow$) | Probing loss (xyz) ($\downarrow$) | Probing loss (rot) ($\downarrow$) | Probing loss (gripper) ($\downarrow$) |
|---|---|---|---|---|
| Ours w/o context | 0.165 | 0.0675 | 0.00861 | 0.928 |
| Ours | **0.152** | **0.0574** | **0.00619** | **0.873** |
| Relative improvement | 7.9% | 15.0% | 28.1% | 5.9% |

Table 7: Zero-shot generalization to an unseen embodiment.

The results from both experiments demonstrate that the embodiment context improves performance and aids generalization to novel embodiments. We hypothesize that while the visual FDM provides general transferability by aligning latent actions with visual changes, the proprio-FDM grounds these latent actions in robot physical dynamics. However, due to data heterogeneity (e.g., different action

definitions / controllers, as discussed previously), the model requires the embodiment context to disambiguate different embodiments and learn a more consistent, grounded latent action space.

## E    LATENT ACTION EXPERT VISUALIZATION

In this experiment, we demonstrate the performance of the latent expert by passing its prediction through the image reconstruction FDM that takes the latent action as inputs and predicts the future observations, which forms a simulated environment for the iteratively executing the latent expert.

Starting from a single initial image, the latent expert and image reconstruction FDM jointly generate different behaviors in videos that follow diverse instructions using only latent actions. We experiment with initial images from RT-1 and Bridge dataset, and show the image clips of generated videos in Figure 9 and Figure 10 with different language instructions. The results show that the latent expert properly follows the language instructions for task solving, where the latent expert properly recognizes the target objects and predict latent actions that move towards the target object.

## F    MORE ABLATIONS ON POLICY MODEL

We primarily conducted ablation studies on two main components: (1) the attention mask and (2) the embodiment context. Our experiments follow the same setting as Table 1 in the main paper. The ablation results below show that both the attention mask and embodiment context are effective in improving performance on two robot platforms: Google Robot and WidowX Robot.

Table 8: Ablation study results for the policy model. The first columns (Pick, Move, Drawer) refer to the Google Robot, and the last columns (Carrot, Eggplant, Spoon, Cube) refer to the WidowX Robot. All numbers are success rates (%).

| Method | Pick | Move | Drawer | Avg. | Carrot | Eggplant | Spoon | Cube | Avg. |
|---|---|---|---|---|---|---|---|---|---|
| Ours | 81.7 | 55.4 | 38.4 | 58.5 | 24.2 | 71.7 | 48.3 | 19.2 | 40.8 |
| Ours w/o mask | 80.3 | 30.6 | 48.8 | 53.2 | 18.3 | 52.5 | 38.3 | 26.7 | 34.0 |
| Ours w/o context | 86.6 | 21.3 | 39.3 | 49.1 | 28.3 | 67.5 | 25.8 | 32.5 | 38.5 |

## G    SIMULATION EVALUATION DETAILS

### G.1    SIMPLER BENCHMARK

We evaluate on all eight SIMPLER (Li et al., 2024e) tasks in the visual matching setting, which include two robot platforms: Google Robot and WidowX.

For Google Robot, the tasks are: (1) pick coke can (including horizontal, vertical and standing can configurations); (2) move an object near a target object; (3) open / close top, middle or bottom drawer; and (4) place apple in a closed drawer, which includes two subtasks: first open top drawer, and then place the apple into the top drawer. On the widowX setup, the tasks consist of: (1) put a carrot on the plate; (2) put an eggplant on the basket; (3) put a spoon on the towel; (4) stack a green cube on a yellow one.

We follow the standard evaluation protocol to test by randomizing both configurations of the environments. For the Google Robot tasks, we execute 300 trials of "Pick Coke Can", 240 of "Move Near", 216 of "Open/Close Drawer", and 108 of "Place Apple in Closed Drawer". For each WidowX task, we use 24 unique configurations. To ensure statistical significance, we test each configuration 10 times, yielding 240 rollouts per task. Reported results (Table 2) are the average success rates across these trials. Please refer to SIMPLER (Li et al., 2024e) for more details.

For a fair comparison, we adopt the published performance metrics for RT-1-X (Collaboration et al., 2023), Octo-base (Octo Model Team et al., 2024), OpenVLA (Kim et al., 2024), RoboVLMs (Li et al., 2024b), MoTo (Chen et al., 2024b), and LAPA (Ye et al., 2024) directly from their respective papers. In the case of GR00T (NVIDIA et al., 2025b), we use the official pretrained checkpoint

and perform fine-tuning on the RT-1/Bridge dataset following the authors' published guidelines accordingly.

# H  LIBERO BENCHMARK

The LIBERO benchmark (Liu et al., 2023a) evaluates knowledge transfer in multitask and lifelong robot learning problems for robotic manipulation, consisting of four task suites: **LIBERO-Spatial** evaluates the model's performance under novel layouts with the same task and object types, **LIBERO-Goal** evaluates the model's performance under novel tasks with the same object types and layouts, **LIBERO-Object** evaluates the model's performance under novel object types with the same tasks and layouts, **LIBERO-Long** evaluates the model's performance under diverse set of objects, layouts and backgrounds. Each task suite contains 10 tasks with 50 human demonstrations per task for fine-tuning.

**Baselines and Experimental Setup**   We compare with the following existing models: Diffusion Policy (Chi et al., 2023) trained from scratch, Octo (Octo Model Team et al., 2024), OpenVLA (Kim et al., 2024), $\pi_0$ (Black et al., 2024), $\pi_0$ FAST (Pertsch et al., 2025), TraceVLA (Zheng et al., 2024) and SpatialVLA  (Qu et al., 2025). For $\pi_0$, we use the open source version (Ren, 2025) and the same training set as our model. All models follow a two-stage pretraining-finetuning protocol. We finetune `villa-X` and `villa-X` w/o latent on the demonstration data of the each task suite separately, and test on the LIBERO simulator for 10 tasks and 20 trials per task on each task suite.

**Experimental Results**   Table 9 summarizes the success rates on each task suite of LIBERO. Our model achieves better performance than existing methods in all the four task suites. Also, our model with latent action achieves higher performance on all the four task suites and average performance, confirming that the proposed latent action expert improves the manipulation performance.

Table 9: Evaluation on 4 LIBERO task suites of `villa-X` and existing methods.

| Method | Spatial | Object | Goal | Long | Average |
|---|---|---|---|---|---|
| Diffusion Policy (Chi et al., 2023) | 78.3 | 92.5 | 68.3 | 50.5 | 72.4 |
| Octo-base (Octo Model Team et al., 2024) | 78.9 | 85.7 | 84.6 | 51.1 | 75.1 |
| OpenVLA (Kim et al., 2024) | 84.7 | 88.4 | 79.2 | 53.7 | 76.5 |
| $\pi_0$ (reimplement (Ren, 2025)) | 88.0 | 88.5 | 87.0 | 61.0 | 81.1 |
| $\pi_0$-FAST (Pertsch et al., 2025) | 96.4 | 96.8 | 88.6 | 60.2 | 85.5 |
| TraceVLA (Zheng et al., 2024) | 84.6 | 85.2 | 75.1 | 54.1 | 74.8 |
| SpatialVLA  (Qu et al., 2025) | 88.2 | 89.9 | 78.6 | 55.5 | 78.1 |
| **Ours** w/o latent | 86.0 | 86.5 | 85.0 | 70.0 | 81.9 |
| **Ours** | **97.5** | **97.0** | **91.5** | **74.5** | **90.1** |

# I  REAL-WORLD ROBOT PLATFORMS EVALUATION DETAILS

## I.1  REALMAN ROBOT ARM

The Realman robot arm setup is shown in Figure 5 (upper). We mount the gripper for Inspire Robot to the Realman RM75 robot arm. We use two camera views, including a primary view camera with the same view point as the images (used to demonstrate different tasks) shown in Figure 5 (upper) and a wrist camera. For fine-tuning of our models, we reinitialize the linear state encoder, action encoder, and action decoder, and tune the full parameters (except for the vision encoder). We fine-tune all the models for 60k gradient steps.

We collect data on the following five tasks with their task instructions:

- Put-in: "Pick the green block from the table into the blue bowl"
- Put-out: "Pick the green block from the blue bowl onto the table"
- Push: "Push the green block to position X" where "X" indicates the nine positions written on the table.

- Stack: "Stack the wooden block onto the green block"
- Unstack: "Unstack the wooden block from the green block"

We collect 375 trajectories (75 trajectories for each task) for fine-tuning. The trajectories are collected at 10Hz. We post-process these trajectories to remove static frames with zero action, resulting in 120 steps on average in one trajectory.

We evaluate the fine-tuned model on seven groups with 10 trials for each group. The first five groups contain the tasks the same as data collection. The last two groups are designed to evaluate the generalization ability of the models. For the "change block color" group, we repeat the previous five tasks but change the green block into blue and red ones. For the "change table cover" group, we change the table cover from red to brown and blue ones.

The visualization example of each task for our model can be found in Figure 11.

### I.2    XHAND DEXTEROUS HAND

The Xhand setup is shown in Figure 5 (lower). The 12-dof Xhand is mounted on a 7-dof XArm robot arm. There are two camera views, including a main 3-rd view camera, and a wrist camera. During fine-tuning, we reinitialize linear encoder and decoder modules for both state and action to accommodate the hand's higher dimensionality.

We use the dataset collected in (Hu et al., 2024) as our finetuning dataset, which comprises roughly 4,000 trajectories spanning 13 task categories and over 50 unique objects. For evaluation, we focus on five representative XHand tasks as depicted in Figure 5, namely pick-and-place, cube stacking, upright cup placement, water pouring, and ball flicking. Each task is assessed under "seen" and "unseen" conditions: in the seen setting, the same objects and backgrounds encountered during training are used, albeit with randomized tabletop positions and optional distractors; in the unseen setting, either the target objects or the scene background (or both) were never encountered during finetuning, totaling more than 20 novel objects. During evaluation, we conducted 50 evaluation runs for the pick-and-place task, 20 runs for cube stacking, and 10 runs for each of the remaining tasks. The visualization example of each task can be found in Figure 12 and Figure 13.

## J    LLM USAGE

Throughout the preparation of this manuscript, we utilized Large Language Models (LLMs) as writing assistants. Their primary roles were to proofread for grammatical accuracy and to help refine the prose.

## K  MORE RESULTS ON PROBING ANALYSIS

In this section, we provide additional results for the experiments described in Section 4.1. Figure 14 presents the probing results under the $L_1$ loss using two complementary visualizations. In the left panel, we display overlapped bar plots where distinct colors represent different methods. In the right panel, we visualize the difference in sample distribution across the same error thresholds.

## L  LATENT PLAN ROLLOUT USING LAM FDM

This section supplements Figure 4 with visualizations using the LAM's image FDM. The procedure involves generating a sequence of latent actions $z_{1:n}$ via the ACT-latent policy, given an initial frame and instruction. We then recursively predict future frames via $\hat{o}_{t+1} = FDM(\hat{o}_t, z_t)$, where the input $\hat{o}_t$ is the output of the last iteration. We observe that the image FDM captures the correct motion intention, although the generated images suffer from severe blurring as errors accumulate during iterative rollouts. This visual degradation justifies our motivation for employing a dedicated world model in Figure 4.

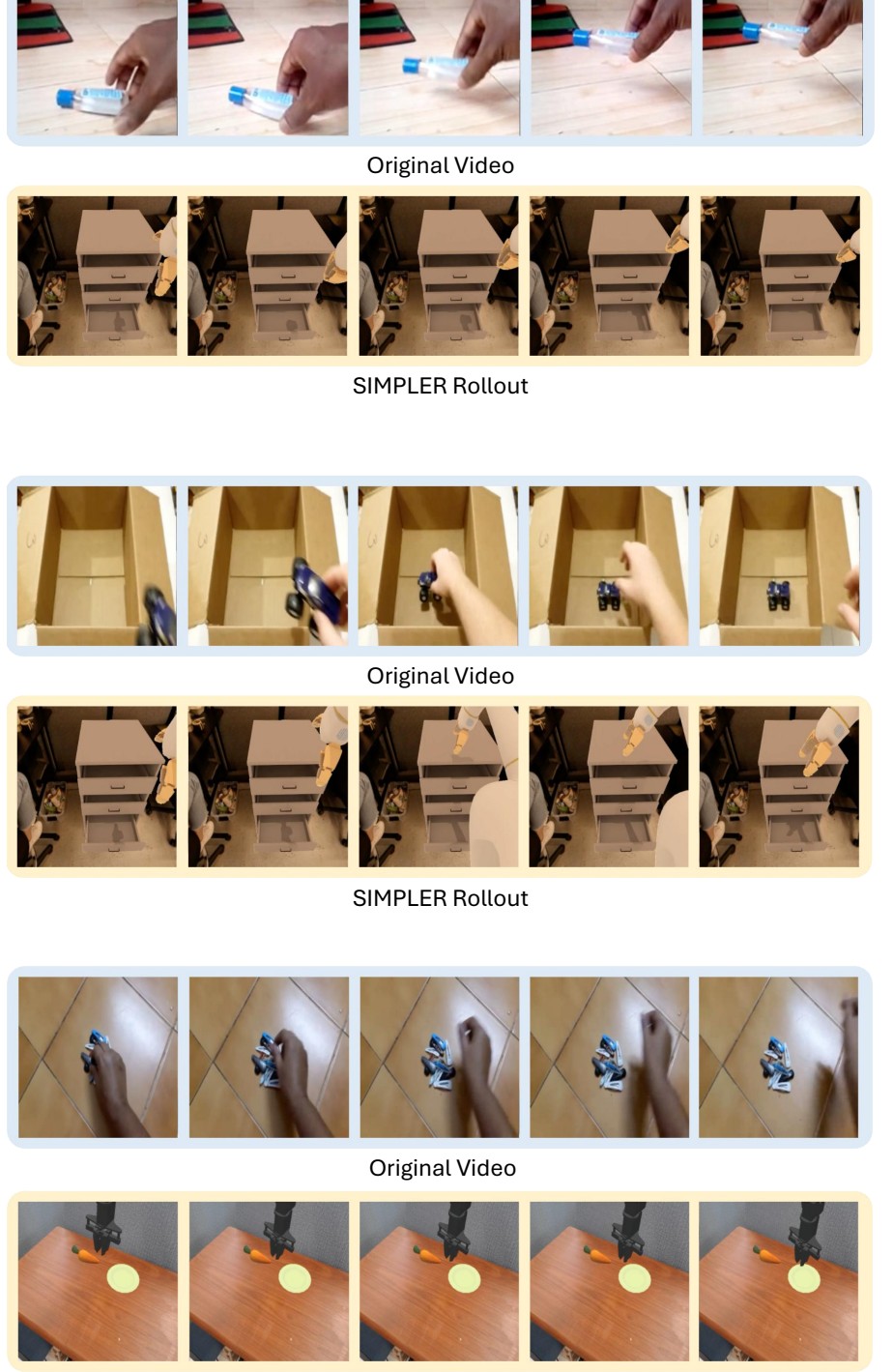

Figure 8: Transfer human video demonstrations into robot actions through `LAM` and proprio FDM in SIMPLER simulator. Upper: the SIMPLER rollout closely reproduce the motion of moving right; Middle: the SIMPLER rollout closely reproduce the motion of moving forward and backward; Bottom: the SIMPLER rollout closely reproduce the motion of moving right.

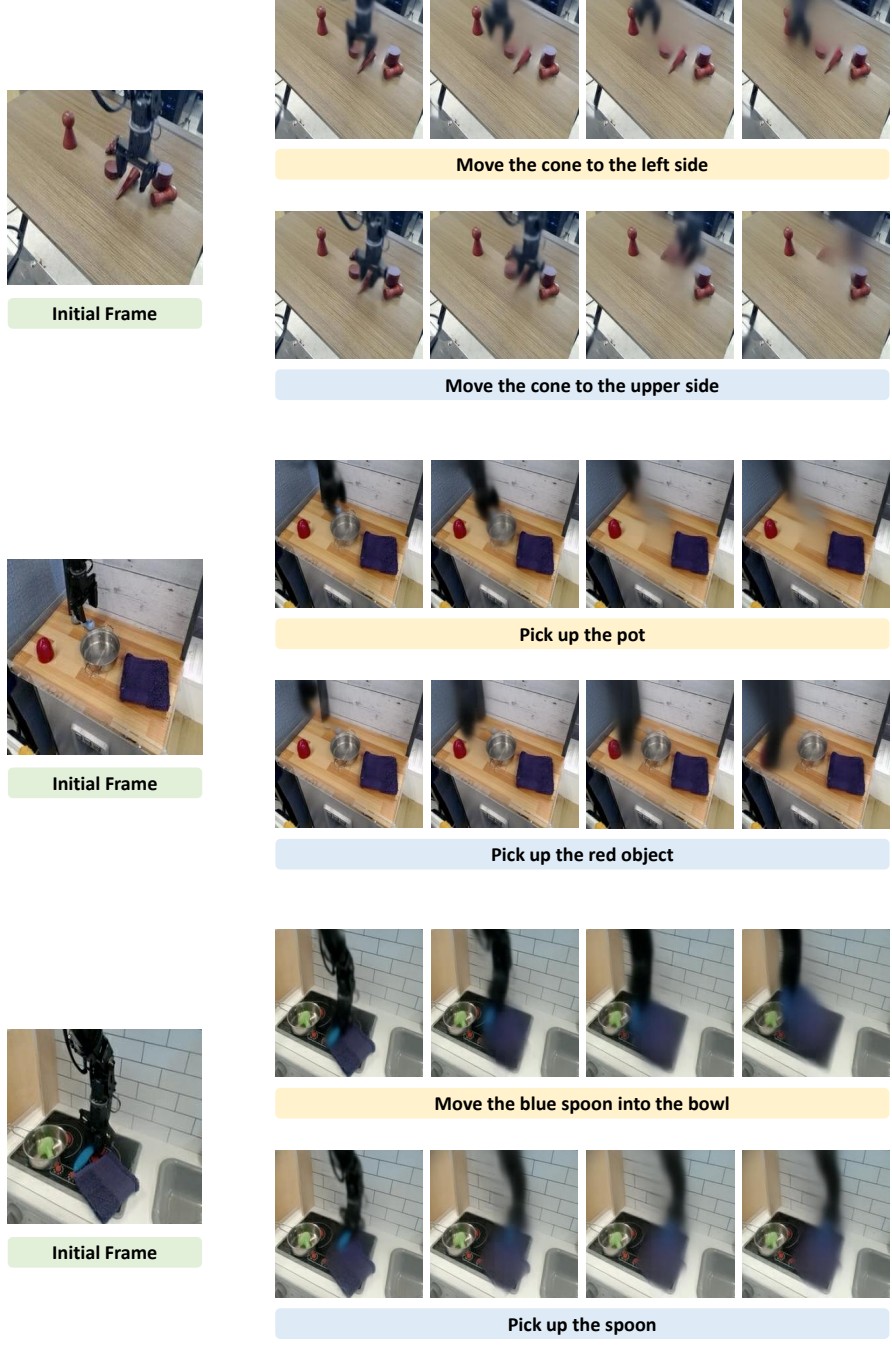

Figure 9: Generated image sequence jointly by the latent expert and the world model via latent actions, following different instructions from the same initial image (Part I).

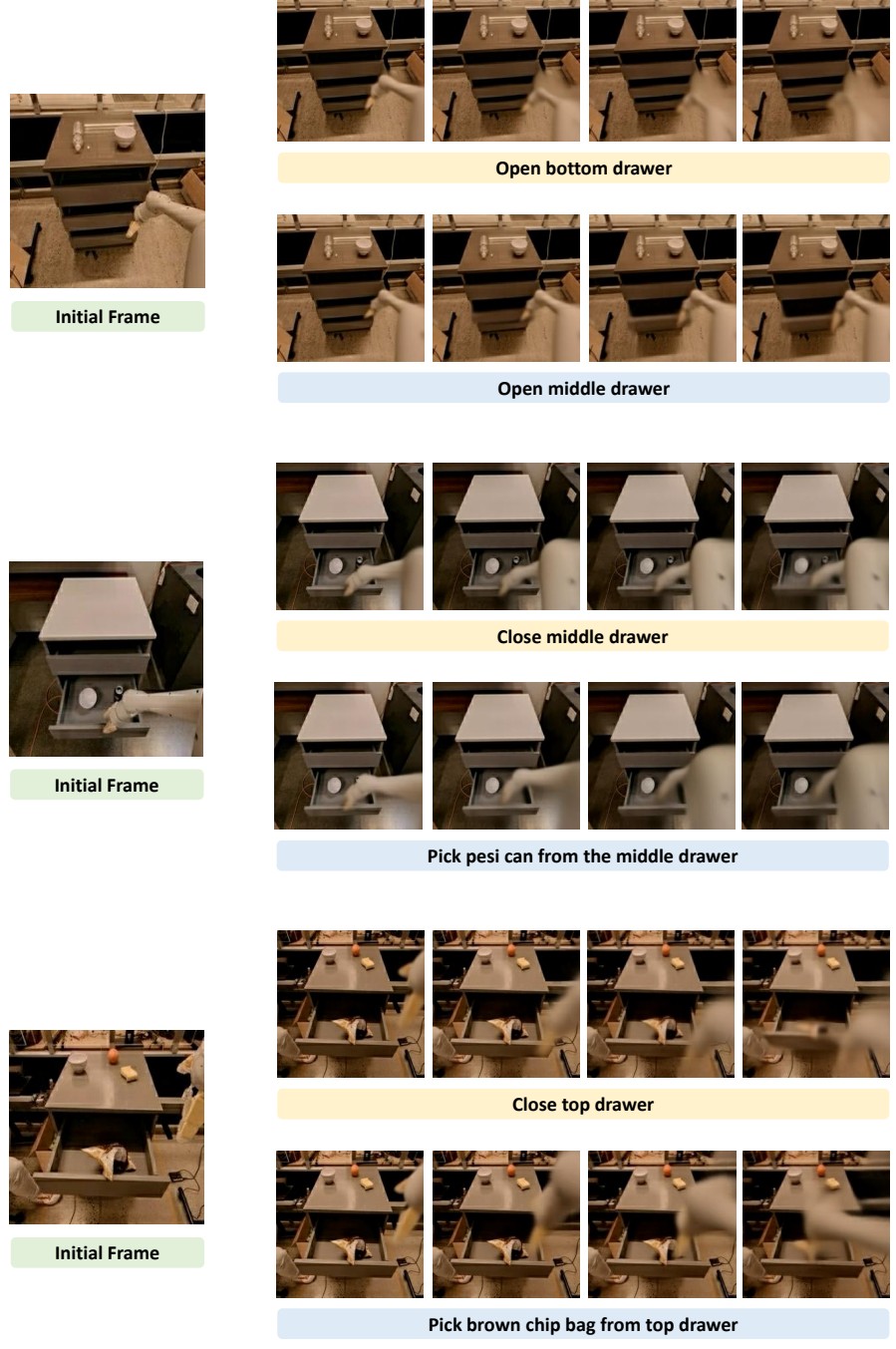

Figure 10: Generated image sequence jointly by the latent expert and the world model via latent actions, following different instructions from the same initial image (Part II).

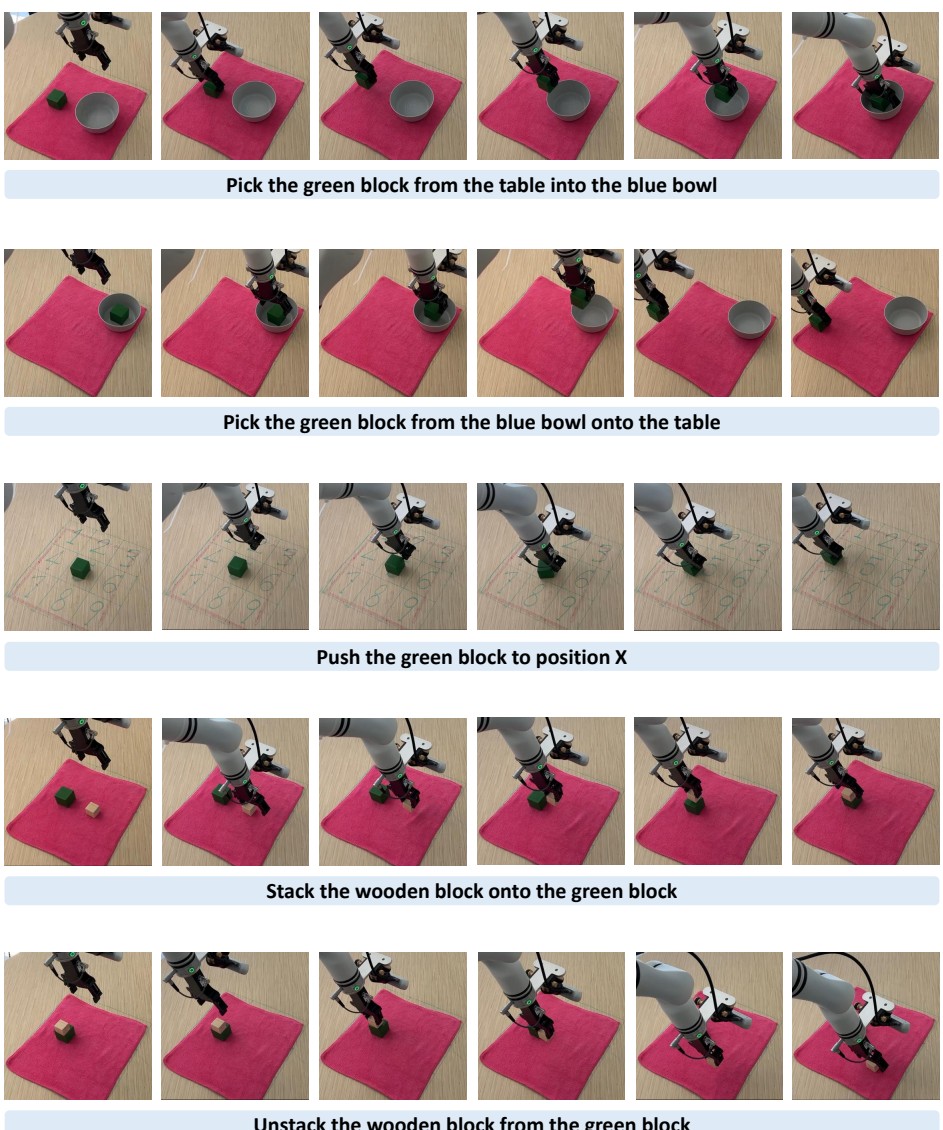

Figure 11: Realman evaluation trajectory examples.

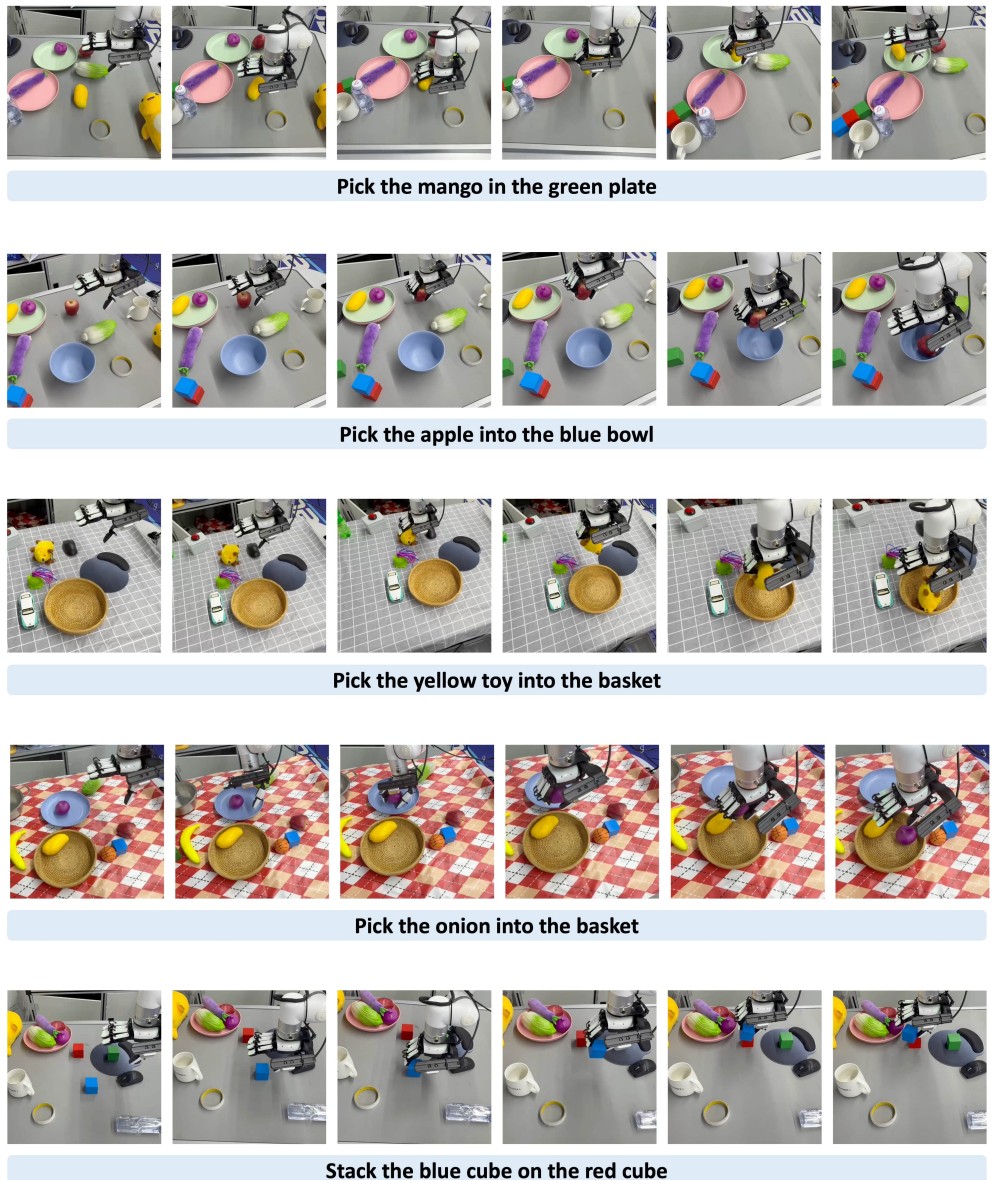

Figure 12: Xhand evaluation trajectory examples (part I).

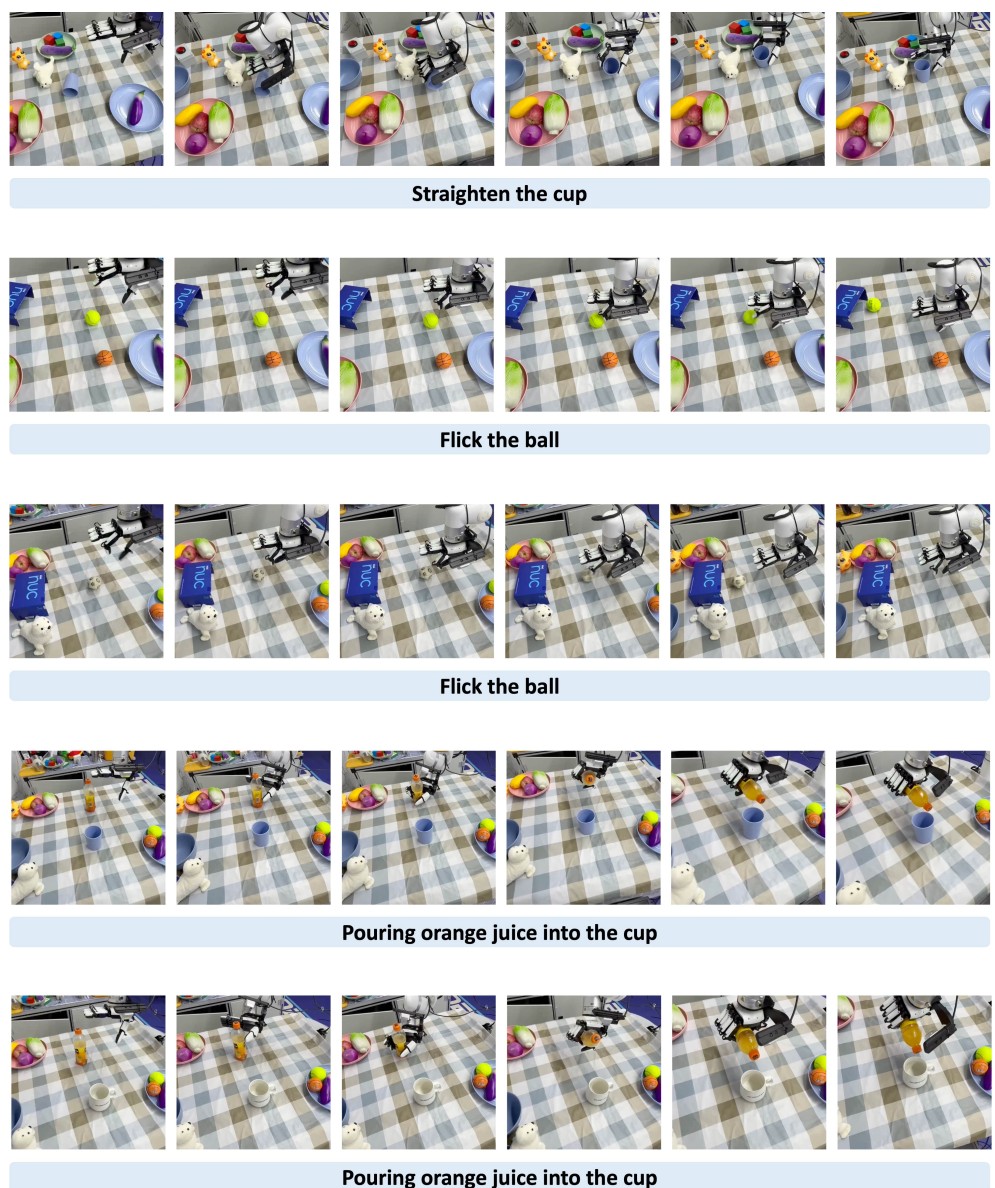

Figure 13: Xhand evaluation trajectory examples (part II).

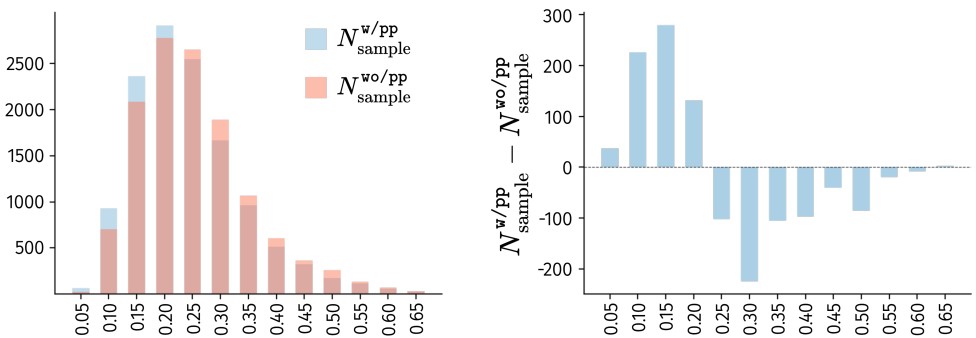

Figure 14: Probing results under $L_1$ loss using different views.

Touch the cookie

Touch the rice

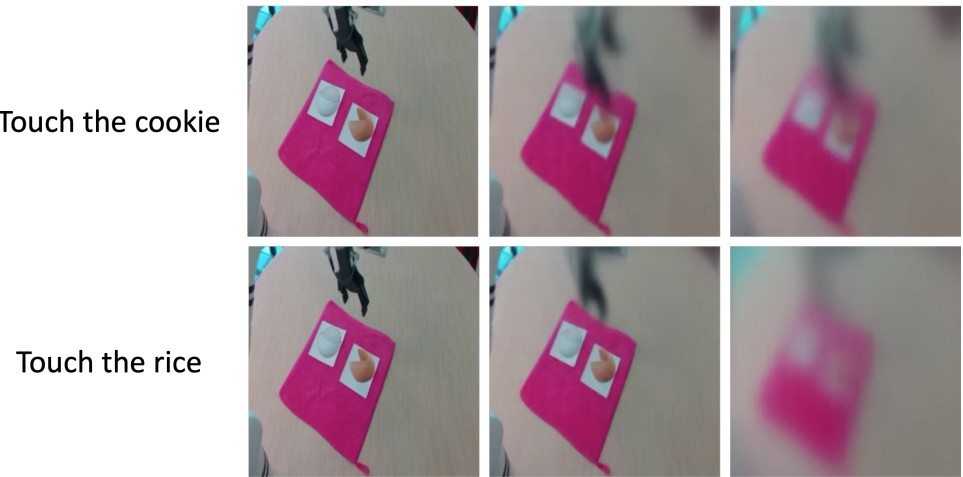

Figure 15: The rollout of latent plan using LAM FDM.

