# OpenReview forum: "villa-X: Enhancing Latent Action Modeling in Vision-Language-Action Models"
_ICLR.cc/2026/Conference — ICLR 2026 Poster_

### Official Review · Reviewer_7xUJ · 2025-10-27

**Soundness:** 3
**Presentation:** 3
**Contribution:** 2
**Rating:** 6
**Confidence:** 4

**Summary:**

This paper presents villa-X (vision-language-latent-action), a framework that integrates latent actions into vision-language-action (VLA) models. The core idea of villa-X consists of two components:
- incorporating existing latent action models (LAMs) into a proprioceptive module to obtain more physically grounded latent actions
- jointly modeling latent actions and real robotic actions.

Experimental results demonstrate that (1) the proprioceptive module enables LAMs to extract higher-quality latent actions, and (2) the resulting VLA model significantly outperforms existing VLA baselines in both simulated and real-world environments.

**Strengths:**

- [S1] villa-X achieves strong performance in both real-world and simulated settings. The authors conduct extensive experiments that convincingly demonstrate the effectiveness of the proposed framework.
- [S2] The paper is clearly written and well-organized. Figures and equations are concise yet informative, effectively illustrating how villa-X operates.

**Weaknesses:**

- [W1] From a high-level perspective, this work largely follows the structure of the existing framework [1]. The process—first training latent action models (LAMs) with a VQ-VAE-style objective to generate pseudo labels (latent actions) for robot data, and then training the VLA model to predict those latent actions—remains similar. The proposed addition of a proprioceptive module and the joint prediction of latent and robotic actions, while useful, appear somewhat incremental.
- [W2] The paper provides limited discussion on why villa-X performs well across diverse environments and tasks. The experiments primarily report performance improvements without deeper analysis of contributing factors or underlying mechanisms.
- [W3] No statistical significance analysis is presented. The reported performance metrics lack measures of variance or confidence, making it difficult to assess the robustness of the claimed improvements.

**References**

[1] Ye et al., Latent Action Pretraining from Videos. In ICLR, 2025.

**Questions:**

- [Q1] In Table 1, the performance of LAPA and Go-1 on the WidowX robot appears noticeably poor. Could the authors elaborate on the underlying cause of this phenomenon?

---

> ### Author Response · Authors · 2025-11-25
> **Response to Reviewer 7xUJ (Part 1)**
>
> Thank you for your positive assessment and encouraging remarks! We are glad that you found our approach to valuable. We have carefully addressed your remaining questions which have helped us clarify our contributions and significantly improve the quality of our manuscript.
>
> ---
>
> **1. Relationship between LAPA**
>
> Thank you for the insightful comment! We acknowledge that our method operates within the broader "Latent Action Learning" paradigm, similar to LAPA [1]，Moto-GPT, UniVLA, and GO-1. While these works share a high-level workflow (pre-training a LAM followed by downstream policy learning), our core contributions address critical limitations in these frameworks:
>
> **Handling Visual Bias for Grounding:** Standard VQ-VAE approaches (like LAPA) often struggle to ground latent actions in physical dynamics. We observed that standard latent spaces are prone to actions causing massive pixel changes (e.g., large arm movements) while ignoring visually subtle but control-critical motions (e.g., gripper rotation or closing). Our introduction of the Proprio-FDM and the corresponding Embodiment Context ($c_e$) fundamentally changes the objective from simple visual reconstruction to dynamics-aware grounding, ensuring these fine-grained physical behaviors are captured.
>
> **Joint vs. Hierarchical:** Unlike previous works (e.g., GO-1, LAPA) that typically treat latent and robot actions hierarchically in separate stages, our Joint Diffusion formulation models them as a single coherent distribution. This enables information flow during the whole denoising process, allowing the policy captures cross-modal dependencies more effectively, which is conceptually distinct and empirically superior.
>
> Our results demonstrate that these specific architectural choices are necessary to achieve the reported gains in success rate and cross-embodiment generalization.
>
>
> ---
>
> **2. Analysis on the performance gain**
>
> Thank you for the constructive advice! We agree that analyzing the underlying mechanisms of performance is as important as the results themselves. We will update the manuscript to include a deeper discussion on the performance gain achieved by Villa-X, focusing on two primary factors:
>
> **Overcoming Visual Bias via Proprioceptive Grounding:** A key factor in our performance is the improved quality of the latent action space. As analyzed in our revised discussion, standard visual-only latent spaces are biased towards actions that cause massive pixel changes (e.g., large arm movements) while often ignoring visually subtle but control-critical motions (e.g., gripper rotation or closing).
>
> Our probing experiments (Section 4.1) demonstrate that adding the Proprio-FDM significantly reduces the $L_{max}$ probing loss on OOD robots. This confirms that our latent actions capture fine-grained physical dynamics that visual losses miss. The ablation study (Table 1) further validates this: removing the Proprio-FDM results in a sharp drop in success rate, proving that this "physical grounding" is the active mechanism driving the performance gain.
>
> **Generalizable Latent Planning in Policy:** Beyond the quality of the latent space itself, our policy demonstrates strong generalization capabilities. As evidenced by the zero-shot rollout visualizations in Figure 4, the policy has internalized a generalizable understanding of physical dynamics from the diverse pre-training data. Furthermore, because the latent space is explicitly grounded in physics (enabled by the Proprio-FDM as discussed above), the ACT-Robot can leverage this high-level planning to generate precise, fine-grained actions for execution.
>
> In summary, the strong performance of Villa-X across diverse environments stems from the synergy between these designs: the Proprio-FDM ensures the representation is physically valid, while the policy architecture ensures this representation is robustly generalized to new contexts.

---

> ### Author Response · Authors · 2025-11-25
> **Response to Reviewer 7xUJ (Part 2)**
>
> **3. Statistical significance in the results**
>
> We appreciate the reviewer's emphasis on statistical rigor. While we reported the aggregated success rates, we ensured the robustness of these metrics by employing a large-scale evaluation protocol, following the standards defined in the simulator benchmark [1].
>
> Specifically, to minimize randomness and ensure the reported means are statistically representative, we averaged results over hundreds of independent rollouts with varying initial configurations for each task. The specific number of evaluation runs for the main results (Table 1 and 2) is detailed below:
>
>
> |             | Pick | Move | Drawer | Carrot | Eggplant | Spoon | Cube |
> |------|------|------|--------|--------|----------|-------|------|
> | Number of Runs | 300  | 240  | 216    | 240    | 240      | 240   | 240  |
>
> This protocol aligns with standard practices in recent literature [2,3,4]. Given the large sample size (N>200 for most tasks), the standard error of the mean is sufficiently low to confirm that the reported performance gaps are robust and not artifacts of random seeding.
>
> [1] Li, X., Hsu, K., Gu, J., Pertsch, K., Mees, O., Walke, H. R., ... & Xiao, T. (2024). Evaluating real-world robot manipulation policies in simulation. arXiv preprint arXiv:2405.05941.
>
> [2] Liu, H., Li, X., Li, P., Liu, M., Wang, D., Liu, J., ... & Zhang, H. (2025). Towards generalist robot policies: What matters in building vision-language-action models.
>
> [3] Chen, Y., Ge, Y., Tang, W., Li, Y., Ge, Y., Ding, M., ... & Liu, X. (2025). Moto: Latent motion token as the bridging language for learning robot manipulation from videos. In Proceedings of the IEEE/CVF International Conference on Computer Vision (pp. 19752-19763).
>
> [4] Yang, J., Tan, R., Wu, Q., Zheng, R., Peng, B., Liang, Y., ... & Gao, J. (2025). Magma: A foundation model for multimodal ai agents. In Proceedings of the Computer Vision and Pattern Recognition Conference (pp. 14203-14214).

---

> ### Author Response · Authors · 2025-11-25
> **Response to Reviewer 7xUJ (Part 3)**
>
> **4. Analysis on LAPA and GO-1**
>
>
> We thank the reviewer for highlighting these results. We attribute the lower performance of these baselines to different factors, and we have updated our analysis based on recent re-evaluations.
>
> **Regarding LAPA:**  In response, we conducted further analysis of the results presented in Table 1 for the WidowX robot.
>
> Specifically, we evaluate two methods (LAPA and our method) on the validation set by reporting the mean L1 loss (L1) over the 7D action space, along with a breakdown into its translational (L1 xyz), rotational (L1 rot), and gripper (L1 gripper) components.
>
> |      | L1(↓)    | L1-xyz(↓) | L1-rot(↓) | L1-gripper(↓) |
> |------|-------|--------|--------|------------|
> | LAPA | 0.244 | 0.236  | 0.268  | 0.196      |
> | Ours | **0.192** | **0.193**  | **0.212**  | **0.131**      |
>
> As shown in the table, the LAPA-style policy exhibits significantly higher L1 error across all dimensions. Visual inspection of the test trajectories corroborates the numerical data. We observed that robots using the LAPA-style approach often exhibit correct high-level intentions (e.g., moving toward the correct object), but frequently fail due to a lack of fine-grained manipulation accuracy (e.g., imprecise grasping).
>
> These findings suggest that the performance gap mainly result in how the policy utilizes latent actions. The LAPA-style two-stage paradigm (Latent Action stage 1 → Robot Action stage 2) appears less effective at preserving representations for high-frequency control details during decoding. In contrast, our Joint Diffusion formulation—which co-denoises latent and robot actions—enforces a tighter coupling between intent and execution, resulting in the superior precision and success rates observed.
>
> **Regarding GO-1:** We would like to provide an update regarding the GO-1 baseline. Our initial submission (prior to the official code release on Sept. 19) approximated GO-1 using an autoregressive loss for latent action prediction.
>
> During the review process, we received feedback suggesting that an L1 loss formulation for latent action might better reflect the intended design, although the official repository and the paper remain ambiguous on this detail. To ensure the most rigorous comparison, we re-implemented and re-evaluated GO-1 using this L1-based formulation. While this update improved the baseline's performance, Villa-X still significantly outperforms the corrected GO-1 baseline (+14.6% average improvement on RT-1 and +4.3% on Bridge)
>
>
> |                      | Pick     | Move     | Drawer   | Avg.     | Carrot   | Eggplant  | Spoon    | Cube     | Avg.     |
> |----------------------|----------|----------|----------|----------|----------|-----------|----------|----------|----------|
> | Ours                 | **81.7** | **55.4** | 38.4     | **58.5** | **24.2** | 71.7      | **48.3** | **19.2** | **40.8** |
> | GO-1-style (L1 loss) | 44.3     | 46.3     | **41.2** | 43.9     | 7.5      | **91.3** | 30.4    | 16.7     | 36.5     |
>
> Despite the improved baseline, we observe that GO-1 yields a significantly higher validation loss on latent actions compared to our method. This performance gap suggests that our joint diffusion modeling is structurally more effective at capturing cross-modal dependencies than GO-1.
>
> ---
>
>
> We hope that our response and the additional experiments have satisfactorily addressed your concerns. We remain fully available to answer any further questions during the discussion period! If you find our response convincing, we kindly ask that you consider re-evaluating the score.

---

### Official Review · Reviewer_oHXE · 2025-10-30

**Soundness:** 3
**Presentation:** 3
**Contribution:** 3
**Rating:** 8
**Confidence:** 4

**Summary:**

This paper introduces a new framework called villa-X, whose core idea is to incorporate latent actions into the pretraining and policy learning of Vision-Language-Action (VLA) models.

**Strengths:**

1. The system design is simple, effective, and scalable.

2. The experiments are sufficient and comprehensive.

3. The writing is clear and provides a good reading experience.

**Weaknesses:**

See questions.

**Questions:**

1. I recently came across latent action learning in a survey paper [1]. How does the latent action learning mentioned in your work differ from that?

2. I also believe that latent learning is a promising approach for solving cross-embodiment transfer. What other potential solutions do you foresee in the future?

3. Equation (3) makes me a bit confused — what exactly is the dataset ID, and why is the context vector ce composed of these two parts?

4. How large is the VILLA-X model in terms of scale?

[1] Towards a Unified Understanding of Robot Manipulation: A Comprehensive Survey. arXiv 2025.

---

> ### Author Response · Authors · 2025-11-25
> **Response to Reviewer oHXE**
>
> Thank you for your positive assessment and encouraging remarks! We are glad that you found our approach valuable. We also appreciate the reference to the survey paper and share your optimism regarding the potential of latent action learning in this field. Below, we address your questions and provide further clarifications.
>
> ---
>
> **1. Regarding the survey**
>
> Thank you for highlighting the comprehensive survey. In the taxonomy presented there, our method falls under the category of Latent Learning for VLA. Our work differs from prior methods discussed in [1] in several key aspects.
>
> **Embodiment-Aware Grounding:** Beyond standard latent learning, we incorporate a Proprio-FDM with an explicit Embodiment Context ($c_e$). This mechanism is critical for handling data heterogeneity while grounding the visual latent actions to the physical dynamics.
>
> **Joint Diffusion Modeling:** Our policy utilizes a joint diffusion policy that co-denoises latent and robot actions. By integrating structured attention masking, this framework achieves a more robust and expressive multi-modal mapping than typical deterministic or decoupled approaches.
>
> We will update Section 2 to discuss these differences in the context of the survey.
>
>
> ---
>
> **2. Future directions for cross-embodiment transfer**
>
> We share the reviewer's optimism regarding latent action learning. Beyond this paradigm, we see several important directions for advancing cross-embodiment transfer:
>
> **Unified 3D Representations:** Current frameworks often rely on 2D images, which require the model to implicitly compensate for viewpoint and camera position differences. Leveraging 3D state representations could enable viewpoint-invariant policies and more effective transfer across embodiments.
>
> **Morphology-Agnostic Representations:** Visual differences in robot appearance could hinder transfer. Learning representations that are agnostic to robot-specific appearance, while retaining task-relevant scene information, could be a promising direction.
>
>
> **Adaptation via Contextual RL:** To handle variations in controllers and physical parameters, contextual RL can enable policies to adapt dynamically to new embodiments by formulating the problem as a contextual MDP, leveraging shared task knowledge while adapting to specific dynamics.
>
>
> ---
>
> **3. Clarification on Equation (3) and the context vector $c_e$**
>
> Thank you for raising this point. The embodiment context $c_e$ is designed to capture both the static and dynamic properties of the robot, which vary significantly across the heterogeneous dataset like OXE mixture. It consists of two concatenated parts:
>
> **Dataset ID (Static):** A learnable embedding vector unique to each dataset. This embedding acts as a proxy for unobserved static properties, such as robot morphology, kinematics, and joint definitions.
>
> **Control Frequency (Dynamic):** A continuous embedding representing the control Hz. We project the scalar frequency value using sinusoidal features followed by an MLP.
>
> By providing these signals, the embodiment context enables the proprio-FDM to disambiguate distinct physical dynamics even when the visual input is similar. This allows the model to learn a consistent, grounded latent action space despite the severe data heterogeneity.
>
> ---
>
> **4. How large is the VILLA-X model in terms of scale?**
>
> We report the model scale based on its two primary components. The Latent Action Model (LAM) consists of 245M trainable parameters, while the downstream Policy model comprises approximately 3B trainable parameters.
>
> ---
>
> We hope our response clarifies the remaining details. We thank you again for your support and look forward to any further discussion

---

### Official Review · Reviewer_XiVL · 2025-10-31

**Soundness:** 3
**Presentation:** 3
**Contribution:** 2
**Rating:** 4
**Confidence:** 4

**Summary:**

The paper introduces VILLA-X, a Vision-Language-Latent-Action framework that advances robot policy learning by improving both how latent actions are learned and integrated into Vision-Language-Action models. It enhances latent action learning through a proprioceptive forward dynamics module that grounds latent representations in physical robot dynamics, and it introduces a joint diffusion-based policy that conditions robot action generation on latent action planning.

**Strengths:**

1. The paper is well-structured and clearly written. The problem motivation is sound, the technical approach is explained logically.

2. The evaluation is thorough, encompassing systematic ablations, major simulation benchmarks (SIMPLER, LIBERO), and real-world deployment on two distinct platforms.

3. The demonstrated capability for zero-shot generalization to novel embodiments addresses a core challenge in the field.

**Weaknesses:**

1. The technical contributions, while valuable, exhibit limited novelty relative to existing literature. The proposed proprioceptive Forward Dynamics Model (proprio-FDM), which grounds latent actions by predicting low-level states, is conceptually similar to the approach of Nikulin et al. [1], who employ a linear decoder on latent tokens to predict actions. The efficacy of this general principle for grounding has also been previously analyzed by Zhang et al. [2]. Furthermore, the architectural design of separate experts for latent and robot actions (ACT-latent and ACT-robot) bears a strong resemblance to the module separation employed in GO-1.

2. The characterization of the GO-1 baseline may be inaccurate. Based on its open-source implementation, GO-1 does not appear to autoregressively predict latent actions using a next-token-prediction (NTP) loss, but rather uses an L1 loss for latent action learning. Consequently, the description in Section 4.2 and the subsequent analysis in Table 1 could be misleading regarding the true nature of this baseline.

3. The experimental comparisons lack benchmarks against several highly relevant contemporary works, notably IGOR [3] and UniVLA [4]. These methods also focus on cross-embodiment generalization and leverage latent actions learned from web videos, making their inclusion critical for properly contextualizing the claimed advancements of this work.

4. The zero-shot generalization analysis in Section 4.3 would be strengthened by clarifying the training procedure for the world model used for visualization.



______

[1] Nikulin, Alexander, et al. "Latent action learning requires supervision in the presence of distractors." arXiv preprint arXiv:2502.00379 (2025).

[2] Zhang, Chuheng, et al. "What Do Latent Action Models Actually Learn?." arXiv preprint arXiv:2506.15691 (2025).

[3] Chen, Xiaoyu, et al. "IGOR: Image-goal representations are the atomic control units for foundation models in embodied ai." arXiv preprint arXiv:2411.00785 (2024).

[4] Bu, Qingwen, et al. "UniVLA: Learning to act anywhere with task-centric latent actions." arXiv preprint arXiv:2505.06111 (2025).

**Questions:**

Please refer to the weaknesses section.

---

> ### Author Response · Authors · 2025-11-25
> **Response to Reviewer XiVL (Part 1)**
>
> We sincerely thank you for their careful reading and insightful suggestions. We found your feedback to be extremely valuable for strengthening the rigor and help improve the quality of our work! We have carefully considered all the questions raised, and our responses are provided below.
>
> ---
>
> **1. Novelty of the proprio-FDM**
>
>
> Thank you for highlighting these relevant works! We have cited and discussed [1] and [2] in Section 2 of our manuscript.
>
>
> Our Proprio-FDM introduces two distinct technical innovations necessitated by our focus on physical grounding and cross-embodiment transfer:
>
> **Forward Dynamics vs. Action Decoding:** Previous works [1,2] typically employ a simple action decoder that maps latent actions directly to action labels. In contrast, our Proprio-FDM models the physical dynamics of the system. It takes the current proprioceptive state and latent action as inputs to explicitly predict the future proprioceptive states alongside the actions. This enforces a stronger physical constraint, ensuring the latent actions are grounded in the robot's actual state transition dynamics rather than just fitting action labels.
>
> **Handling Heterogeneity via Embodiment Context:** Uniquely, we address the challenges present in large-scale, cross-embodiment robot datasets such as those in OXE, which exhibit significant heterogeneity (e.g., varying controller parameters, control frequencies, and hardware). This necessitates one of the novel technical contributions here: the embodiment context $c_e$ as described in Section 3.1. This component allows the proprio-FDM to disambiguate between different embodiments, enabling the learning of a consistent, grounded latent action space despite physical variations.
>
> To demonstrate that this is a non-trivial contribution essential for performance, we conducted an ablation study comparing our full method ("Ours") against a version without the embodiment context ("Ours w/o context"). Both were trained on 10% of the OXE dataset.
>
> **(1) Performance on validation dataset:** We measured the reconstruction loss of image FDM and proprio FDM on the validation set. The inclusion of the embodiment context significantly reduces both image FDM error and proprio-FDM loss.
>
>
> | Method               | Image FDM loss (↓) | Proprio FDM loss (↓) |
> |----------------------|--------------------|----------------------|
> | Ours w/o context     | 0.068              | 0.078                |
> | Ours                 | **0.057**          | **0.070**            |
> | Relative improvement | 16.2%              | 10.3%                |
>
> **(2) Zero-Shot Generalization to a Novel Embodiment:** We evaluated zero-shot transfer to a completely unseen Realman robot arm (as described in Section 4.4). We performed the action probing experiment (Section 4.1), training an MLP to predict robot actions from the fixed latent actions inferred by the IDM.
>
>
> | Method               | Probing loss (↓) | Probing loss (xyz) (↓) | Probing loss (rot) (↓) | Probing loss (gripper) (↓) |
> |----------------------|------------------|------------------------|------------------------|----------------------------|
> | Ours w/o context     | 0.165            | 0.0675                 | 0.00861                | 0.928                      |
> | **Ours**             | **0.152**        | **0.0574**             | **0.00619**            | **0.873**                  |
> | Relative improvement | 7.9%             | 15.0%                  | 28.1%                  | 5.9%                       |
>
> These results confirm that simply applying a prediction head is insufficient for heterogeneous datasets. The embodiment context is critical for grounding latent actions in robot physical dynamics, yielding a 28.1% improvement in rotation modeling on unseen robots. This adaptation extends the general principles established in prior work to address the heterogeneity inherent in large-scale embodied AI.
>
> [1] Nikulin, Alexander, et al. "Latent action learning requires supervision in the presence of distractors." arXiv preprint arXiv:2502.00379 (2025).
>
> [2] Zhang, Chuheng, et al. "What Do Latent Action Models Actually Learn?." arXiv preprint arXiv:2506.15691 (2025).

---

> ### Author Response · Authors · 2025-11-25
> **Response to Reviewer XiVL (Part 2)**
>
> **2. Discussion and Implementation with GO-1**
>
> Thank you for the clarification regarding the GO-1 implementation details!
>
> **Regarding the GO-1 baseline implementation:** We agree on the importance of accurate baselines. We would like to clarify that at the time of our submission, the official GO-1 code had not yet been released (the repository was made public on Sept. 19, the same day as the ICLR abstract deadline), and the original paper did not specify the loss function details. Consequently, we originally approximated it with an autoregressive loss. Furthermore, upon inspecting the recently released codebase, we were unable to locate the loss function for latent action part during tuning (we only found the robot action loss). However, acknowledging your feedback regarding the L1 loss, we have re-implemented the GO-1 baseline using the L1 loss on latent actions.
>
> **Experimental Results:** As shown in the updated table below, while the L1 loss improves the baseline, Villa-X still outperforms the corrected GO-1 baseline: +14.6% average improvement on RT-1 and +4.3% on Bridge.
>
> |                      | Pick     | Move     | Drawer   | Avg.     | Carrot   | Eggplant  | Spoon    | Cube     | Avg.     |
> |----------------------|----------|----------|----------|----------|----------|-----------|----------|----------|----------|
> | Ours                 | **81.7** | **55.4** | 38.4     | **58.5** | **24.2** | 71.7      | **48.3** | **19.2** | **40.8** |
> | GO-1-style (L1 loss) | 44.3     | 46.3     | **41.2** | 43.9     | 7.5      | **91.3** | 30.4    | 16.7     | 36.5     |
>
> **Architecture & Novelty:** This persistent performance gap serves as empirical validation of our architectural contributions. We observed that Villa-X achieves significantly lower evaluation loss than GO-1 for both latent and robot action prediction, especially for latent action. This result demonstrates that our specific design choices, including joint diffusion modeling, attention masking, etc., can capture cross-modal dependencies more effectively, and are critical for performance.

---

> ### Author Response · Authors · 2025-11-25
> **Response to Reviewer XiVL (Part 3)**
>
> **3. New baselines: IGOR / UniVLA**
>
> We thank the reviewer for suggesting these highly relevant baselines. We agree that comparing against state-of-the-art methods like IGOR [1] and UniVLA [2] is important for contextualizing our advancements.
>
> To study the cross-embodiment generalization ability, we integrated the LAM modules from baselines into our standardized policy framework (in Section 4.2) and trained them under identical conditions (10% Bridge RT-1 data and 100% SSv2 data).
>
> For IGOR [1]: We re-implemented the LAM architecture following the details provided in their paper. For UniVLA [2]: We utilized their officially open-sourced LAM weights and architecture.
>
> As shown in the table below, our method achieves the highest average success rate across the tasks (+2.3% vs. UniVLA on RT-1, +2.0% on Bridge).
>
>
> |        | Pick     | Move     | Drawer | Avg.     | Carrot   | Eggplant | Spoon    | Cube     | Avg.     |
> |--------|----------|----------|--------|----------|----------|----------|----------|----------|----------|
> | Ours   | **81.7** | 55.4 | 38.4   | **58.5** | 24.2 | **71.7**     | **48.3** | 19.2 | **40.8** |
> | IGOR   |   62.6    |     60.4     |   45.3    |   56.1       |       **33.3**   |    68.3      |    44.1      |    9.1      |    38.7      |
> | UniVLA |  51.7    |   **61.6**     |   **55.5**     |   56.2      |      27.5    |  69.1        |    37.5      |   **21.2**       |  38.8        |
>
> While UniVLA and IGOR excel in specific object manipulations (e.g., Drawer or Carrot), our method demonstrates more robust performance across the full distribution of tasks. This suggests that our Proprio-FDM and Embodiment Context provide a more stable and generalizable grounding for latent actions compared to the objectives used in IGOR and UniVLA.
>
>
> [1] Chen, Xiaoyu, et al. "IGOR: Image-goal representations are the atomic control units for foundation models in embodied ai." arXiv preprint arXiv:2411.00785 (2024).
>
> [2] Bu, Qingwen, et al. "UniVLA: Learning to act anywhere with task-centric latent actions." arXiv preprint arXiv:2505.06111 (2025).
>
>
>
> ---
>
> **4. world model training procedure**
>
> Thank you for the suggestion! We will add a detailed description of the world model training in the revised manuscript.
>
> **Training Procedure:** We employ a continuous-time Rectified Flow model trained to predict future frames $o_{t+1:T}$ conditioned on previous observation and the sequence of future latent actions $z_{t:T-1}$. To leverage strong video priors, we initialize our model with the pre-trained OpenSora checkpoint [1]. Crucially, we repurpose the original text conditioning mechanism by replacing the text embeddings with our latent action embeddings and reinitialized the related parameters. The fine-tuned world model is conditioned solely on history frames and the latent action plan; it does not have access to the text instruction. Consequently, the latent action sequence must inherently encapsulate the necessary semantic information to control the video generation.
>
>
> **Inference/Rollout:** During evaluation, the rollout follows a two-step process:
> (1)Given an initial frame and a text instruction, the policy generates a sequence of latent actions.
> (2)The World Model then takes these generated latent actions (and the history frames) as input to generate the future video trajectory.
>
>
> [1] Zheng, Z., Peng, X., Yang, T., Shen, C., Li, S., Liu, H., ... & You, Y. (2024). Open-sora: Democratizing efficient video production for all. arXiv preprint arXiv:2412.20404.
>
> ---
>
> We believe that the revisions and new experimental results directly address the issues raised. We are happy to engage in further discussion if any points remain unclear. We hope these improvements warrant a positive re-evaluation of our work!

---

### Official Review · Reviewer_kiPs · 2025-10-31

**Soundness:** 3
**Presentation:** 3
**Contribution:** 3
**Rating:** 6
**Confidence:** 3

**Summary:**

The paper presents an approach to learn latent actions, and leverage latent actions when predicting robot actions through joint diffusion. In addition to learning forward and inverse dynamics models, as is done in existing work, the paper proposes to learn an embodiment-conditioned model which predicts future robot states and actions. During inference-time, the action head predicts both latent actions and robot actions given image observations, language instructions, proprioceptive states and embodiment embedding, where an attention mask is used to enforce the factorization of the conditional probability.

**Strengths:**

- The proposed approach has strong empirical results. In particular, in Table 2, the proposed approach outperforms state-of-the-art VLA models, such as OpenVLA, OpenVLA-OFT, $\pi_0$, and Gr00T. It also outperforms other algorithms such as MoTo and LAPA.

- Using an embodiment-specific embedding is an interesting approach to leveraging diverse robot datasets, where different datasets have slightly different dynamics and action spaces. Ablation study shows that the embodiment embedding has a positive impact on the success rate.

- The ablation study is in general quite thorough, showing the effectiveness of each component of the proposed approach, e.g., latent action model, proprioceptive state prediction, etc.

**Weaknesses:**

- Although the main design components are validated through ablation studies. Some finer-grained design choices lack discussion and study. See more details in the **Questions** section.

- Figure 3, probing experiment results, is not very easy to understand. Perhaps a plot showing the distribution of error in different intervals, for both w/ pp and w/o pp,  would be more informative. It is also not very convincing why $L_\infty$ (max across all dimensions) is preferred over $L_1$ (summing/averaging over all dimensions).

- The results in 4.3, although helpful and intuitive, are not very informative scientifically. Since the images are obtained from a “separately trained world model”, it is hard to tell whether the latent action actually encodes the robot behavior or these outcome images just come from hallucination from the world model. It may be good to swap it with another ablation study in the appendix.

- It is unclear why only Gr00t is used as the baseline for real-world experiments, where other state-of-the-art VLA models are left out.

- The tasks evaluated in the experiment are mainly simple pick and place tasks, without much dexterity.

- Minor: Incomplete sentence in Appendix D.3: "Both models were trained on 10..."

**Questions:**

- How is $K$, the number of future steps when training FDM and IDM, determined?

- Why is the prediction only for $o_t$ and $o_{t+K}$ when training observation F/IDM in Equation (1), but for the entire sequence when prediction robot states $q_{t+1:t+K}$ and actions $a_{t+1:t+K}$ in Equation (2)?

- In Equation (3), why is the context vector conditioned on dataset ID instead of robot ID. Wouldn’t it make sense for two datasets with the same robot to share an ID?

- Why does the action head predict $(n-1)K$ latent actions and $m$ robot actions? How to choose $n$ and $m$?

- Why is the robot action branch “overly relying on latent actions” harmful? Isn’t the latent action supposed to provide sufficient information for action prediction (Equation (2))?

- What is the action space for the model? How does the gripper command translate to Xhand command in the realworld experiments?

- In Appendix A.1, how is the codebook size of $32$ decided?

- In Appendix B, can you explain how different $\tau$ distribution can be used for latent actions and robot actions? This seems contradictory to (5) where all actions are bundled together during denoising.

- In Appendix F, is the attention mask ablation referring to the block-wise causal attention mask or random masking of attention during training?

- In Appendix H, why is OpenVLA-OFT not included in Table 9? Especially considering that it is included in Table 2.

---

> ### Author Response · Authors · 2025-11-25
> **Response to Reviewer kiPs (Part 1)**
>
> Thank you for your positive assessment and encouraging remarks! We are glad that you found our approach to be valuable. We are very appreciate for the careful reading and detailed comments, which have helped us clarify our contributions and significantly improve the quality of our manuscript. We have addressed your remaining questions to further solidify the manuscript.
>
> ---
>
> **1. For Figure 3, why use L_max? Why not use overlapping bars?**
>
> Regarding the metric choice, we utilized $L_{max}$ (maximum deviation) instead of $L_1$ because in robotic control, task success is often determined by the worst-case error in any single dimension (e.g., one misaligned joint can cause a grasp failure). Therefore, $L_{max}$ serves as a stricter and more practically relevant metric for this domain.
>
> However, we agree that visualizing the error distribution is highly informative. As suggested, we have added the corresponding $L_1$ probing results and the overlapped bar plots in Figure 14, Appendix K. These additional visualizations provide a more comprehensive view of the performance improvements.
>
> ---
>
> **2. about the world model's results in Section 4.3**
>
> Thank you for raising this important concern. We clarify that, given an intial frame and text instruction, we first use the pretrained policy to generate latent action sequence. Then, the world model generates future frames conditioned **only on** history image frames and the generated latent action sequence; it **does not** have access to the text instruction. Therefore, for the world model to generate a rollout that aligns with a specific text instruction (as seen in Section 4.3), the necessary semantic information must be contained within the latent action sequence itself. If the latent actions did not encode the behavior, the world model would have no signal to generate the task-specific outcome.
>
> To further validate this and rule out hallucination, we have included visualizations of the Forward Dynamics Model (FDM) rollouts in Figure 15, Appendix L. These visualizations are generated iteratively using the FDM in latent action model without passing through the separate world model. While these direct FDM outputs are naturally blurrier, they demonstrate clear intention for task completion. This confirms that the structure of the movement is determined by the policy, and the world model serves only to render these valid trajectories in higher fidelity.
>
> ---
>
> **3. more baselines in real world experiments?**
>
> We have conducted additional real-world experiments with more baselines in response to your suggestion. Within the constraints of the rebuttal timeline and computational resources, we prioritized the most relevant comparisons and the results are presented below
>
>
> | Model                     | Pick-in | Pick-out | Stack | Unstack | Push | Change block color | Change table cover |
> |---------------------------|---------|----------|-------|---------|------|--------------|------------|
> | Open-Pi-0                | 40      | 70       | 50    | 70      | **60**   | 40           | 40         |
> | GR00T                    | 30      | 70       | 10    | 60      | 10   | 50           | 30         |
> | OpenVLA-OFT              | **50**      | 70       | 50    | 80      | 10   | 40           | 50         |
> | Ours (w/o latent action) | 40      | 80       | **60**    | 70      | 30   | 40           | 30         |
> | Ours                     | 30      | **100**      | 50    | **100**     | 50   | **60**           | **60**         |
>
> ---
>
> **4. evaluation on tasks other than pick-and-place?**
>
> We appreciate this feedback and agree that evaluating dexterity is crucial. We would like to highlight that we tested on several dexterous tasks using the dexterous hand platform in our manuscript, as shown in Appendix I.2, Figure 13. These tasks go beyond simple pick-and-place:
> - "Flick the ball": This requires dynamic, non-prehensile manipulation where the finger must perform a preparatory circular motion before releasing to strike the target ball.
> - "Pour orange juice": This involves stable grasping coupled with precise, continuous orientation control to pour liquid.
> - "Straighten the cup": This requires object reorientation and handling objects with complex affordances (e.g., cylindrical bodies), similar to the pouring task.
>
> These results can evaluate the fine-grained control ability and show the dexterity of our method beyond static grasping.

---

> ### Author Response · Authors · 2025-11-25
> **Response to Reviewer kiPs (Part 2)**
>
> **5. How is $K$ determined?**
>
> The hyperparameter $K$ determines the time interval between frames. The selection of $K$ involves a trade-off: if $K$ is too small, the visual change is negligible. If $K$ is too large, the visual change will be excessive and even the scene will change.
>
> To ensure consistent action granularity across diverse datasets, we calibrate $K$ based on the raw control frequency of each dataset. For robot datasets, we select $K$ such that one latent action corresponds to $K$ robot actions, where $K$ is rounded up so that the effective latent action frequency is approximately 2–3 Hz (e.g., for a robot operating at 5 Hz, we set $K=2$). For human videos, where motion is typically faster, we empirically set $K$ to correspond to a frequency of 5 Hz. This approach ensures that latent actions encode motion of a comparable magnitude across all domains, facilitating transfer and generalization.
>
>
>
> ---
>
> **6. Why only use image $o_{t+K}$ but use actions $a_{t:t+K}$?**
>
> The decision is based on the trade-off between computational efficiency and information density.
>
> Since images are high-dimensional data, predicting the full sequence of intermediate frames $o_{t:t+K}$ would significantly increase the computational burden and training memory requirements. In contrast, the action space is low-dimensional, so predicting the full sequence $a_{t:t+K}$ adds negligible overhead while capturing the necessary high-frequency control signals. Our experiments show that predicting the final state $o_{t+K}$ provides good results empirically. We leave predicting all images to future research.
>
> ---
>
> **7. Why use dataset ID instead of robot ID?**
>
> We initially considered using Robot ID, but found it insufficient to capture all domain differences. For instance, multiple datasets may use the same arm, but with different grippers, different underlying control parameters, or different camera setups. Using the Dataset ID allows us to uniquely identify these specific configurations. We also agree that incorporating Robot ID could help and we view it as a promising direction for future work.
>
> ---
>
> **8. Latent action chunk size $n$ vs robot action chunk size $m$**
>
> We use action chunking for both the high-level latent policy and the low-level robot policy. To ensure the high-level plan covers a sufficiently long period, we tune the parameters such that $nK>m$. We enforce this inequality ($>$ instead of $=$) to account for varying control frequencies across diverse datasets. Since perfect temporal alignment is difficult to guarantee across different domains, a longer latent horizon acts as a safety buffer. In practice, we use a latent chunk size of $n=6$ and a robot action chunk size of $m=4$.

---

> ### Author Response · Authors · 2025-11-25
> **Response to Reviewer kiPs (Part 3)**
>
> **9. Why is overly relying on latent action harmful? What is the attention mask in App.F?**
>
> Clarifying the role of latent actions is crucial. While latent actions capture the high-level plan and physical dynamics (facilitated by the auxiliary loss in Eq. (2)), they are effectively coarse motions and are not sufficient for fine-grained low-level control on their own.
>
> The "harm" of over-reliance stems from a shortcut learning phenomenon. In our framework, the final action is derived hierarchically (use simplified notation here, please refer to Eq. (4) in the paper):
>
> $\pi = \pi_{latent}(z_{t:t+n}|o_t,l)\pi_{robot}(a_{t:t+m}|o_t,l,q_t,c_e,z_{t:t+n})$
>
> We observed that without the attention mask, the Robot Expert $\pi_{robot}$ learns to ignore the other inputs including visual image, language instruction and proprioceptive state, shortcutting directly to the latent action $z$. This results in poor generalization and robustness. This occurs because the robot becomes "blind" to the current task, merely decoding the latent plan, which is not sufficient to accomplish the task by its nature. When the predicted latent plan is even slightly imperfect during rollout, the robot actions becomes even worse.
>
> To address this, we employ the random attention mask described in Section 3.2 (randomly mask out the attention from robot actions to the latent actions). This forces the Robot Expert to attend to both the high-level intent (z) and the immediate sensorimotor context. This ensures the policy utilizes the latent action as a high-level guide while relying on vision and proprioception for precise execution.
>
> Our ablation study (Appendix F) confirms that this mechanism is critical for performance. As shown in the table below, removing the attention mask leads to a significant drop in success rates across tasks (e.g., a 5.3% drop in the RT-1 benchmark and 6.8% in the Bridge benchmark), validating that preventing over-reliance on latent actions is essential for robustness.
>
>
> | Method        | Pick | Move | Drawer | Avg. | Carrot | Eggplant | Spoon | Cube | Avg. |
> |---------------|------|------|--------|------|--------|----------|-------|------|------|
> | Ours          | **81.7** | **55.4** | 38.4   | **58.5** | **24.2**   | **71.7**     | **48.3**  | 19.2 | **40.8** |
> | Ours w/o Mask | 80.3 | 30.6 | **48.8**   | 53.2 | 18.3   | 52.5     | 38.3  | **26.7** | 34.0 |
>
>
> ---
>
> **10. What is the action space?**
>
> The action space for the gripper robot arm is 7 dimensions, including 3d for xyz, 3d for euler rotation and 1d for gripper openness. The action space for the X-Hand is 18 dimensions, including 3d for xyz, 3d for euler rotation, and 12d for finger joints.
>
> As detailed in Appendix B, we adopt a variant of HPT [1] style for policy learning to handle the difference in action space. Specifically, we employ separate MLP adapters (action/state encoders and decoders) for each dataset to project different physical embodiments into a shared latent space. Therefore, regarding the real-world XHand experiments, because our architecture supports variable dimensions, we simply initialize new dedicated action adapters specifically matching the XHand's action dimension during fine-tuning. This allows the model to learn to output XHand control signals directly from the shared policy representation.
>
>
>
> [1] Wang, L., Chen, X., Zhao, J., & He, K. (2024). Scaling proprioceptive-visual learning with heterogeneous pre-trained transformers. Advances in neural information processing systems, 37, 124420-124450.

---

> ### Author Response · Authors · 2025-11-25
> **Response to Reviewer kiPs (Part 4)**
>
> **11. How is the codebook size 32 decided?**
>
> We determined the codebook related hyperparameters (codebook size and the number of codes) empirically. We found that if the overall space is too large, the latent actions across different datasets do not overlap sufficienty, which hinders transfer performance. Therefore, we selected a codebook size of 32 with 2 codes in total to balance expressivity and generalization.
>
>
> ---
>
> **12. Noise sampling technique in $\tau$?**
>
> Thank you for the careful reading! We simplified the notation in Equation (5) for readability. But we will elaborate the procedure in details below and we will update the manuscript accordingly.
>
> We employ two distinct Beta distributions, where the distribution for latent actions is skewed towards higher noise levels. To ensure the diffusion process remains temporally consistent, we use a quantile matching strategy. Specifically, we first sample
> $\tau_r$ for the robot actions and then calculate $\tau_l=ICDF_l(CDF_r(\tau_r))$. $ICDF_l$ is the inverse CDF of the latent action distribution. This guarantees that the relative diffusion progress remains synchronized.
>
> The rationale behind this design parallels that of attention masking as discussed in Response 9. By subjecting the latent actions to a higher noise level, we effectively prevent the policy from learning shortcuts. Empirically, we observe that this specific noise schedule leads to better evaluation performance.
>
> ---
>
>
> **13. Additional baseline scores in Table 9?**
>
> We thank the reviewer for pointing this out. We will update Table 9 in Appendix H to include the results.
>
> ---
>
> **14. Misc: Typos**
>
> We thank the reviewer for their careful reading and attention to detail. We have corrected these typos in the revised manuscript to improve the presentation quality.
>
> ---
>
> We hope that our response and the additional experiments have satisfactorily addressed your concerns. We remain fully available to answer any further questions during the discussion period! If you find our response convincing, we kindly ask that you consider re-evaluating the score.

---

### Author Response · Authors · 2025-12-04
**Summary of Contributions and Rebuttal Updates for Paper 10970**

Dear Area Chair,

Thank you for overseeing the review process. We would like to provide a brief factual summary of our contributions and the updates made during the rebuttal period.

**Summary of Contributions:**


`Villa-X` advances the latent-action paradigm by improving both how latent actions are learned and how they are incorporated into VLA pre-training. Applying these new techniques enables the model to generate latent action plans in a zero-shot fashion, even for unseen embodiments and open-vocabulary symbolic understanding. These techniques enable `Villa-X` to achieve superior performance across diverse simulation tasks in SIMPLER and on two real-world robotic setups involving both gripper and dexterous hand manipulation.


**Reviewer Consensus:**

We are encouraged that reviewers recognized the value of this work, highlighting:

- Strong empirical results and thorough evaluation (kiPs, 7xUJ, XiVL, oHXE).
- Zero-shot generalization results, which addresses a core challenge in the field (XiVL).
- Simple, effective, and scalable system design (oHXE).
- Clear writing and presentation (XiVL, oHXE, 7xUJ).

**Rebuttal Updates:**

In our responses, we have addressed all reviewer questions. A summary of the key updates follows:


- **Additional Baselines & Comparisons: (kiPs,XiVL)** We added new baselines in both real-world and simulation settings (responding to kiPs and XiVL). Crucially, we re-implemented the GO-1 baseline using the exact loss function suggested by reviewer XiVL, as the original implementation was not publicly accessible. The results show that Villa-X consistently outperforms these updated baselines.
- **Novelty Clarification (XiVL,7XUJ):** We clarified the distinct novelty of Villa-X. we showed that Villa-X advances beyond the previous latent-action paradigm by effectively mitigating visual bias for grounding through proprio-FDM in LAM and capturing cross-modal dependencies through joint-diffusion in policy. We explicitly contrasted our Proprio-FDM with standard action-decoders (a key concern of XiVL), showing that explicitly modeling forward dynamics with a novel design on embodiment context is crucial for success on large-scale, cross-embodiment datasets. We validated this via ablation studies showing removing our key design significantly degraded performance. Additionally, we highlighted how our joint-diffusion mechanism captures cross-modal dependencies better than prior approaches.
- **Extended Experimental Analysis(kiPs,7XUJ):** We provided a more detailed analysis of the results, including new visualizations, performance metrics, and a deeper explanation of the performance gains achieved by our method compared to baselines.
- **Methodological Details (kiPs,oHXE):** We expanded our explanation of the model design, offering comprehensive details on the implementation, training procedure, and the rationale behind our hyperparameter selection.
- **Clarification on Statistical Significance (7XUJ):** We confirmed that our simulation evaluation adheres to common standards. We conducted between 216 and 300 repetitions per configuration to minimize statistical uncertainty.
- **Clarification on Task Complexity (kiPs):** We emphasized that our evaluation suite extends beyond simple pick-and-place tasks to include complex dexterous manipulation on a multi-fingered hand platform, such as pouring water, ball flicking, and object straightening.
- **Clarification on Zero-Shot Generalization (kiPs):** We detailed how our latent action learning enables the VLM to generate effective plans for unseen embodiments and novel objects (e.g., unseen emoji cards) without additional training.


We appreciate the reviewers’ engagement and the opportunity to clarify our contributions. We believe the discussion reflects a clear consensus that Villa-X is technically sound and provides significant empirical advancements in latent action modeling and policy learning.

Please let us know if any additional information would be helpful!

Best regards,

Authors

---

### Meta-Review · Area_Chair_tbHJ · 2026-01-07

**Summary:**

This work receives 8664 at initial review.  Three reviewers (XiVL, oHXE, 7xUJ) raise concerns about the similarity with existing latent action learning methods. After rebuttal, the authors give discussions and comparisons with exisitng latent action models.  Thus, I recommend acceptance.

**Reviewer Concerns:**

Concerns about the similarity with existing latent action learning methods are mostly solved.

**Reviewer Scores:**

The reivewer XiVL might change his score to positive since questions regarding the related works are answered.

---

### Decision · Program_Chairs · 2026-01-26

Accept (Poster)